# Pattern-Guided Adaptive Prior for Structure Learning

**Lyuzhou Chen, Yijia Sun, Yanze Gao, Xiangyu Wang**[*],
**Derui Lyu, Taiyu Ban, Xin Wang, Xiren Zhou, Huanhuan Chen**[*]
School of Computer Science and Technology, University of Science and Technology of China
96 Jinzhai Rd, Hefei, China, 230026
`{clz31415@mail., sunyijia2002@mail., gaoyz@mail., sa312@,}`
`{drlv@mail., banty@mail., wz520@mail., zhou0612@, hchen@}ustc.edu.cn`

## Abstract

Learning the causality between variables, known as DAG structure learning, is critical yet challenging due to issues such as insufficient data and noise. While prior knowledge can improve the learning process and refine the DAG structure, incorporating prior knowledge is not without pitfalls. In particular, we find that the gap between the imprecise prior knowledge and the exact weights modeled by existing methods may result in deviation in edge weights. Such deviation can subsequently cause significant inaccuracies when learning the DAG structure. This paper addresses this challenge by providing a theoretical analysis of the impact of deviation in edge weights during the optimization process of structure learning. We identify two special graph patterns that arise due to the deviation and show that their occurrence increases as the degree of deviation grows. Building on this analysis, we propose the **P**attern-**G**uided **A**daptive **P**rior (PGAP) framework. PGAP detects these patterns as structural signals during optimization and adaptively adjusts the structure learning process to counteract the identified weight deviation, thereby improving the integration of prior knowledge. Experiments verify the effectiveness and robustness of the proposed method.

## 1 Introduction

Discovering directed acyclic graph (DAG) structures of variable associations from observational data, also known as structure learning [Scanagatta et al., 2019, Heinze-Deml et al., 2018], is gaining significant attention in many practical applications [Peters et al., 2017, Gong et al., 2022]. However, the broad application of structure learning is frequently hindered when relying on data alone. Such difficulties arise primarily from challenges like insufficient data and unavoidable noise [Ramsey et al., 2018]. To overcome these limitations and improve practical applicability, prior knowledge is often incorporated. This knowledge, typically derived from domain expertise about the existence of specific relationships [Constantinou et al., 2023], allows for the integration of established insights. By doing so, prior knowledge can guide the DAG's structure, reduce the search space, and thereby enhance the efficiency and reliability of the learning process [Eggeling et al., 2019, Chen et al., 2023].

In practical scenarios, the prior knowledge available from domain experts is often imprecise, since it typically indicates the existence of dependencies between variables rather than specifying their exact magnitude or detailed functional form[1] [Kitson et al., 2023]. This imprecision contrasts with the requirements of existing modeling approaches, which often describe the relationships within a DAG structure using precise numerical weights for the connections between variables [Zheng et al., 2018]. Indeed, most current methods widely incorporate such imprecise prior knowledge, yet the potential

---

[1]This paper refers to priors that correctly assert the existence of a directional relationship (e.g., variable A influences variable B) as *accurate*, but not *precise* because they do not specify an exact numerical strength.

39th Conference on Neural Information Processing Systems (NeurIPS 2025).

risks and problems arising from the inherent gap between the non-specific nature of these priors and the precise parameter demands of models have not been fully explored. This direct introduction of imprecise priors often means that existing methods face difficulties in the reconciliation process, potentially leading to suboptimal learning outcomes.

One critical consequence of this issue is what we term "deviation". Because imprecise prior knowledge typically lacks exact numerical specification for relationship strengths, forcibly applying constraints based on it can compel learned edge weights to diverge from their true underlying values. Such deviation has two critical properties. Firstly, the misspecification of one edge's weight due to deviation can propagate and interfere with the accurate learning of other parameters within models. Secondly, the extent of this adverse impact can increase with the magnitude of the initial deviation.

Despite the potential for such deviations, existing prior-based structure learning methods primarily aim to align the learned graph with given prior knowledge. These methods often overlook the risk that forcing adherence to imprecise priors can cause edge weight deviations that can distort the learned structure. Generally, approaches to integrate priors range from more rigid enforcement (e.g., compelling edge existence and specific weight ranges) to more flexible techniques (e.g., using penalty terms to encourage adherence)[2]. Both types of methods can inadvertently lead to deviations. Rigid enforcement can result in weights unrepresentative of the true data-generating process if the constraints sharply conflict with data evidence or true relationship strengths. Similarly, flexible methods can also induce deviations by disproportionately emphasizing prior conformance over data fidelity, ultimately affecting the learned structure.

To address this challenge, this paper first theoretically explores the impact of edge weight deviation on structure learning within the widely-used linear Structural Equation Model (SEM) framework. We demonstrate that such deviation gives rise to two kinds of graph patterns. The prevalence of these patterns increases with the degree of deviation. To assess their association with the deviation, we quantify their expected frequency. This analysis shows they are likely markers of deviation-induced errors rather than naturally occurring structures. Therefore, leveraging these patterns as structural signals of edge weight deviation during the optimization process, this paper proposes the **P**attern-**G**uided **A**daptive **P**rior (PGAP) framework for learning reasonable edge weights. Our main contributions are outlined below.

- A comprehensive theoretical exploration of how imprecise priors cause edge weight deviation and impact continuous optimization in structure learning.
- The formal identification and analysis of two graph patterns manifesting as reliable structural signals of such edge weight deviation.
- The development of the PGAP framework, which leverages these patterns as structural signals to adaptively adjust the learning process, mitigating erroneous structures.

## 2 Related Work

Structure learning aims to discover variable associations from observational data. Traditional approaches include constraint-based methods relying on conditional independence tests [Spirtes et al., 2001, Colombo et al., 2012], score-based methods evaluating graph fit [Chickering, 2002], and hybrid techniques [Tsamardinos et al., 2006]. A significant recent advancement frames structure learning as a continuous optimization problem [Zheng et al., 2018], enabling the use of gradient-based algorithms and spurring developments like NOTEARS, GOLEM [Ng et al., 2020], DAGMA [Bello et al., 2022], and various extensions addressing non-linear relationships or using different optimization strategies [Wei et al., 2020, Yu et al., 2019, Lachapelle et al., 2019, Deng et al., 2023].

Prior knowledge is often used to improve learning accuracy, especially with limited data [Constantinou et al., 2023]. In traditional combinatorial search, priors may be enforced via hard constraints (e.g., requiring/forbidding specific edges or ancestral relationships [De Campos and Castellano, 2007, De Campos et al., 2009, Chen et al., 2016]) or used as soft constraints to guide the search process or scoring [Castelo and Siebes, 2000, Borboudakis and Tsamardinos, 2014, Eggeling et al., 2019]. Similar strategies have been adapted for continuous optimization frameworks. Hard constraints can

---

[2]These two main strategies are often termed "hard" and "soft" prior integration methods, respectively. A detailed discussion of these approaches can be found in Section 2.

restrict edge weight parameters to specific ranges [Sun et al., 2023, Chen and Ge, 2022], while soft constraints typically add prior-based penalty terms to the objective function [Yang et al., 2023, Wang et al., 2024] or directly manipulate gradients during optimization [Bello et al., 2022].

However, existing methods for incorporating prior knowledge in continuous optimization often focus primarily on ensuring the resulting graph qualitatively satisfies the given constraints. They may overlook a crucial side effect: enforcing a qualitative prior (e.g., edge existence) within a quantitative model can lead to deviations in the enforced edge's weight, which, as we show, can propagate and distort the learned weights of other edges. Addressing this challenge is the central motivation for our work. A more detailed discussion of related work can be found in Appendix A.

## 3 Preliminaries

**Notation**   This paper employs standard matrix and vector notation. Subscripts typically indicate specific elements, column vectors, or selections based on index sets. For example, given a matrix $A$, $A_{ij}$ denotes the element in the $i$-th row and $j$-th column, and $A_j$ may denote the $j$-th column vector. For a subset of indices $S \subseteq \{1, \ldots, d\}$, $X_S$ denotes the subvector $(X_i)_{i \in S}$.

**Bayesian Network**   Consider a Bayesian network defined over a set of random variables $X = (X_1, \ldots, X_d)$. Its structure is represented by a DAG $G = (V, E)$, where $V = \{X_1, \ldots, X_d\}$ denotes the set of nodes[3], and $E$ denotes the set of edges. The joint distribution of $X$, denoted as $P(X)$, is Markov with respect to DAG $G$, which means $P(X) = \prod_{i=1}^{d} P(X_i \mid X_{\text{Pa}_i})$, where $\text{Pa}_i$ represents the set of parent nodes of node $X_i$ in graph $G$.

**Structural Equation Model**   This paper considers the case where the random variables follow a linear structural equation model (SEM), described by:

$$X = B^T X + \epsilon \tag{1}$$

Here, $B \in \mathbb{R}^{d \times d}$ represents the weighted adjacency matrix of graph $G$, with each non-zero element $B_{ij}$ indicating an edge $(X_j, X_i)$ in $E$ and describing the linear dependency. The vector $\epsilon = (\epsilon_1, \ldots, \epsilon_d)^T$ denotes the noise, which is assumed to have zero mean and possess a diagonal covariance matrix $\Omega = \text{diag}(\sigma_1^2, \ldots, \sigma_d^2)$.

**Structure Learning**   The primary objective of structure learning is to estimate the DAG structure $G$ (characterized by an estimated adjacency matrix $W$ during estimation) by analyzing an $n \times d$ data matrix $\mathbf{X}$, where each row corresponds to an independent and identically distributed sample $X$ from the joint distribution $P(X)$. This task is often formulated as a constrained optimization problem:

$$\min_{W \in \mathbb{R}^{d \times d}} Q(W; \mathbf{X}) \quad \text{subject to} \quad G(W) \in \text{DAGs} \tag{2}$$

Here, $Q$ is a scoring function that assesses the fit between the data matrix $\mathbf{X}$ and the graph $G(W)$ corresponding to the candidate adjacency matrix $W$. Common scoring functions for linear SEMs include least squares [Loh and Bühlmann, 2014] and negative log-likelihood [Bühlmann et al., 2014]:

$$Q_{ls}(W; \mathbf{X}) = \frac{1}{2n} \|\mathbf{X}W - \mathbf{X}\|_F^2$$

$$Q_{nll}(W; \mathbf{X}) = \frac{1}{2n} \sum_{i=1}^{d} \log \|\mathbf{X}W_i - \mathbf{X}_i\|_2^2 \tag{3}$$

**Continuous Optimization**   Recently, [Zheng et al., 2018] introduced the NOTEARS method, which employs a continuously differentiable function $h(W)$ to characterize the acyclicity of the graph $G(W)$. This transforms the combinatorial constraint $G(W) \in \text{DAGs}$ into a smooth equality constraint, enabling continuous optimization techniques:

$$\min_{W \in \mathbb{R}^{d \times d}} Q(W; \mathbf{X}) \quad \text{subject to} \quad h(W) = 0$$

$$h(W) := \text{tr}(e^{W \odot W}) - d \tag{4}$$

---

[3] We do not distinguish between nodes and random variables, as this simplification does not lead to confusion.

Here, $\odot$ denotes element-wise multiplication and $e^{W \odot W}$ is the matrix exponential. It has been shown that $G(W)$ is a DAG if and only if $h(W) = 0$. Fulfilling this condition ensures that the solution to the optimization problem corresponds to a DAG structure. Following NOTEARS, subsequent research has introduced similar functions to encapsulate the acyclicity of a DAG and further incorporate properties that benefit the optimization process, such as the work of [Ng et al., 2020] and [Bello et al., 2022].

**Prior Knowledge**    In practical scenarios, structure learning can be enhanced by incorporating prior knowledge about the system being modeled, often available from domain experts or previous studies. This incorporation helps resolve ambiguities within the Markov equivalence class, reduces the search space, and improves learning accuracy, especially with limited or noisy data.

This paper defines prior knowledge as $\pi = (X_i \to X_j)$, which indicates the existence of a directed edge from variable $X_i$ to $X_j$ in the true DAG $G$. To ensure prior knowledge benefits structure learning, most studies assume its accuracy (i.e., that the provided prior knowledge aligns with edges in the true DAG $G$). However, even accurate prior knowledge requires careful integration to avoid unintended negative consequences, particularly concerning learned edge weights.

## 4    Pattern-Guided Adaptive Prior Framework for Structural Learning

This section unfolds our approach. First, we theoretically analyze how the integration of imprecise priors can lead to edge weight deviation and subsequently impact the structure learning process. Second, we explore how this deviation manifests as two specific structural patterns within the learned graph. Finally, we detail the PGAP framework, which leverages these patterns as structural signals to adaptively adjust the learning parameters in a two-stage process. Please refer to Appendix B for the proof of the mentioned lemma.

### 4.1    Impact of Edge Weight Deviation

In this section, we detail our principal findings concerning the impact of edge weight deviation on structure learning in SEM continuous optimization, showing that enforcing prior knowledge introduces deviations that interfere with learning other weights.

Our theory is based on the interaction of deviations with other edge weights during optimization of the scoring function. Therefore, we will first analyze the properties of the scoring function. In fact, the optimization of the scoring function

$$W^* = \arg \min_W Q(W; \mathbf{X}) \tag{5}$$

can decompose into independent sub-problems for each variable $X_j$, resembling ordinary least squares (OLS) regression:

$$W_j^* = \arg \min_{W_j} \left( \mathbf{X} W_j - \mathbf{X}_j \right)^2 \tag{6}$$

This conclusion is true for common scoring functions like least squares ($Q_{ls}$) and negative log-likelihood ($Q_{nll}$). Since the variables in DAG have a topological order, for each $X_j$, we only need to consider the edge pointing to it from its predecessors:

$$W_{t_j,j}^* = \arg \min_{W_{t_j,j}} \left( \mathbf{X}_{t_j} W_{t_j,j} - \mathbf{X}_j \right)^2 \tag{7}$$

Here, $t_j$ is the set of topological predecessor nodes to $X_j$ in the graph defined by $W$. In particular, if the set of predecessors $t_j$ aligns with the predecessors in the true DAG, $t_j^*$, (e.g., if the variables $X_1, X_2, \ldots, X_d$ are ordered following the true DAG, then $t_j^* = \{1, 2, \ldots, j-1\}$), the accurate edge weights can be determined directly from sufficient data, as stated in the Lemma 1:

**Lemma 1.** *Given a node $X_j$, assume that $t_j^*$ represents the set of predecessors of $X_j$ consistent with an order of the true DAG $G$. The regression coefficient obtained by linearly regressing $X_j$ on $t_j^*$ corresponds directly to the relevant element in the adjacency matrix $B$ of the true DAG, i.e.:*

$$B_{t_j^*,j} = \arg \min_{W_{t_j^*,j}} \left( \mathbf{X}_{t_j^*} W_{t_j^*,j} - \mathbf{X}_j \right)^2 \tag{8}$$

Now, consider the influence of prior knowledge integration. Assume a prior knowledge $(X_\beta \to X_j)$ is incorporated. Let $W^*_{\beta,j}$ be the optimal weight for edge $(X_\beta, X_j)$ obtained from the OLS regression of $X_j$ onto its current predecessors $t_j$ (Eq. 6). If the prior-influenced result (defined as $\hat{W}_{\beta,j}$) differs from this data-optimal weight, we define the deviation as $\gamma = \hat{W}_{\beta,j} - W^*_{\beta,j}$. This deviation $\gamma$ represents the discrepancy introduced by imposing the prior on the quantitative SEM weight. This occurs, for example, when methods enforcing this prior either fix the weight $W_{\beta,j}$ (a hard constraint) or add penalties to strongly push $W_{\beta,j}$ towards non-zero values (a soft constraint).

The crucial question is how this deviation $\gamma$ on the prior edge $(X_\beta \to X_j)$ affects the learning of other weights, specifically the weights $W_{k,j}$ for other incoming edges $(X_k, X_j)$ where $k \in t_j - \beta$[4]. We analyze this by considering the optimization problem where $\gamma$ is fixed (abstracting the effect of prior)[5]. The fix of $\gamma$ implies the fix of $\hat{W}_{\beta,j}$, so we only need to optimize the weights $W_{t_j-\beta,j}$:

$$\hat{W}_{t_j-\beta,j} = \arg\min_{W_{t_j-\beta,j}} \left( \mathbf{X}_{t_j-\beta} W_{t_j-\beta,j} + \mathbf{X}_\beta \hat{W}_{\beta,j} - \mathbf{X}_j \right)^2 \tag{9}$$

Let $W^*_{t_j-\beta,j}$ denote the optimal weights for non-prior edges obtained from the original OLS problem (Eq. 6) involving all predecessors $t_j$. The following lemma quantifies how the deviation $\gamma$ propagates to affect these other weights.

**Lemma 2.** *Let $\gamma = \hat{W}_{\beta,j} - W^*_{\beta,j}$ represent the deviation of the prior-influenced weight $\hat{W}_{\beta,j}$ (derived from (6)) from the optimal OLS weight $W^*_{\beta,j}$ obtained when regressing $X_j$ onto the set of predecessors $t_j$ (derived from (9)). The difference between the prior-influenced weights for other incoming edges $W^*_{t_j-\beta,j}$ and $\hat{W}_{t_j-\beta,j}$ is a linear function of $\gamma$, expressed as:*

$$\hat{W}_{t_j-\beta,j} - W^*_{t_j-\beta,j} = \frac{\gamma}{\theta_{\beta,\beta}} \theta_{t_j-\beta,\beta} \tag{10}$$

*where $\theta$ is the inverse of the covariance matrix of the variable set $t_j$.*

We further consider scenarios where the predecessors $t_j$ align with the true predecessors $t_j^*$. In this situation, $\beta \in \{1, 2, \ldots, j-1\}$, and $\theta_{t_j-\beta,\beta}$ can be expressed as:

$$\theta_{k\beta} = \begin{cases} -\frac{B_{k,\beta}}{\sigma_\beta^2} + \sum_{i=\beta+1}^{j-1} \frac{B_{\beta,i} B_{k,i}}{\sigma_i^2} & k < \beta \\ -\frac{B_{\beta,k}}{\sigma_k^2} + \sum_{i=k+1}^{j-1} \frac{B_{\beta,i} B_{k,i}}{\sigma_i^2} & \beta \le k < j \end{cases} \tag{11}$$

Equation (11) shows explicitly how the impact term $\theta_{k,\beta}$ depends on the true graph structure $B$ (via direct edges like $B_{k,\beta}$ or $B_{\beta,k}$ and common children structures) and the noise characteristics $\Omega$ (via $\sigma^2$ terms). Combined with Lemma 2, this illustrates how deviation $\gamma$ on edge $(X_\beta, X_j)$ propagates to affect edge $(X_k, X_j)$ based on the underlying true graph pathways connecting $X_k$ and $X_\beta$.

Lemma 1 and Lemma 2 elaborate on the dual role of prior knowledge. Firstly, the prior knowledge can qualitatively guide optimization to better structural approximations by helping identify true predecessors $t_j^*$. However, if the prior knowledge introduces a quantitative deviation $\gamma = \hat{W}_{\beta,j} - W^*_{\beta,j}$, this deviation can propagate and affect the learned weights of other edges $(X_k, X_j)$ for $k \in t_j - \beta$, potentially leading to the generation of spurious edges or incorrect weights, even when the set of predecessors $t_j$ correctly matches the true predecessors $t_j^*$. This underscores the importance of judiciously using prior knowledge, considering not just its qualitative correctness but also its quantitative impact.

**Remark 1.** In the above analysis, we have not discussed the acyclicity constraint of the optimization problem (4). Although the acyclic constraint of DAG may cause differences between the real optimization and linear regression, the introduction of the constraint $h(W) = 0$ imposes a topological order among the variables, ensuring that successor nodes do not point to predecessor nodes. During the continuous optimization process, the constraint $h(W) = 0$ must be met (or approached), meaning the learned weight matrix $W$ always corresponds to a graph structure possessing an order. Consequently, even with the acyclicity constraint, the optimization problem to estimate the incoming edges for node $X_j$ still resembles a linear regression.

---

[4]We use $t_j - \beta$ to represent the deletion of element $\beta$ from set $t_j$.

[5]Note that although $\gamma$ changes during the learning process, analyzing its impact for a given value allows us to understand the characteristics of the resulting DAG structure.

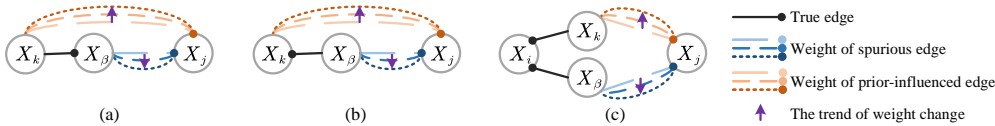

Figure 1: (a,b) If the edge $(X_k, X_\beta)$ or $(X_\beta, X_k)$ exists in the true graph, then as $\gamma$ increases, the learned weight $W_{k,j}$ will gradually deviate from true weight $B_{k,j}$. (c) When there is a common child node $X_i$ between $X_\beta$ and $X_k$, as $\gamma$ increases, the difference between $W_{k,j}$ and $B_{k,j}$ will increase.

## 4.2 Deviation-Induced Graph Patterns Formation

In this section, we show that weight deviation introduces spurious edges, disrupting the learned DAG. This motivates the use of graph patterns as deviation indicators.

Based on the findings from Lemma 2, the impact of the weight difference $\gamma$ on structure learning is linear. Setting aside the coefficient $\theta_{\beta,\beta}$ for simplicity, we concentrate on the coefficient $\theta_{k,\beta}$. Assuming the noise variance as a positive constant, we explore the conditions within the weighted adjacency matrix $B$ where $\theta_{k,\beta} \neq 0$:

**Lemma 3.** *Assuming $t_j = \{1, 2, \ldots, j-1\}$, let $\theta$ be the inverse of the submatrix of the covariance of X corresponding to the variable set $t_j$. Then the condition under which $\theta_{k,\beta}$ is non-zero is given by: $B_{k,\beta} \neq 0$ or $X_\beta$ and $X_k$ have a common child node $X_i$, when $k < \beta$; $B_{\beta,k} \neq 0$ or $X_\beta$ and $X_k$ have a common child node $X_i$, when $k > \beta$.*

It should be noted that a Lebesgue zero-measure set case is ignored in the Lemma 3, which refers to the case where $X_\beta$ and $X_k$ share a common child $X_i$, and there exists a specific non-zero value $B_{k,\beta}$ such that $\theta_{k,\beta} = 0$. Essentially, assuming this special case does not occur is equivalent to the faithfulness assumption [Koller, 2009, Spirtes et al., 2001].

When a node $X_k$ meets the conditions specified in Lemma 3, the weight deviation $\gamma$ will alter the solution to the optimization problem (9). In particular, if $\gamma$ is significant, it might cause the edge weight $W_{k,j}$ obtained in the optimization problem (6) to deviate from zero in optimization problem (9). This means that even if $B_{k,j} = 0$ and all the true predecessor nodes of node $j$ are known, $\gamma$ can still induce the formation of new edges. Figure 1 visually illustrates how weight deviation can lead to the creation of extra edges. As can be seen in Figure 1, the creation of extra edges gives rise to two kinds of graph patterns. In cases (a) and (b), a triangular pattern is formed within the DAG corresponding to the learned adjacency matrix $W$. This structure involves two nodes associated with the prior knowledge ($X_\beta \rightarrow X_j$) and a third node $X_k$ that meets the conditions in Lemma 3. In case (c), $X_\beta$, $X_k$, and $X_j$ form a collision structure, and simultaneously, $X_\beta$, $X_k$, and $X_i$ constitute another collision structure. This case is referred to as a "double collision pattern".

Our focus on these two patterns is not an arbitrary choice but is theoretically grounded. Lemma 3 precisely characterizes the necessary and sufficient conditions for a deviation on a prior-enforced edge to propagate and induce a single spurious edge. The two conditions identified in the lemma, a direct connection or a common child path between the two parent nodes, correspond exactly to the formation of the triangular and double collision patterns, respectively. This demonstrates that these patterns are the exhaustive structural manifestations of this fundamental error mechanism. Therefore, they are not arbitrary heuristics but are leveraged as the principled, theoretically-derived indicators of prior-induced error in our framework.

As $\gamma$ increases, these extra edges and the associated patterns become more prevalent. To quantify the potential prevalence of these deviation-induced patterns compared to naturally occurring ones, we analyze their expected frequency using the standard Erdős–Rényi random graph model $G(d, p)$, where $d$ is the number of nodes and $p$ is the probability of an edge existing between any pair of nodes (Analysis of general models is possible but produces complex results that obscure intuition). This analysis helps justify using these patterns as indicators, but the underlying mechanism linking deviation to these patterns (Lemmas 2 and 3) is based on general SEM properties and graph structures, not limited to ER graphs. Indeed, the conditions in Lemma 3 involve local structures (direct connections or common children paths), implying these patterns may appear in any topology (such as scale-free networks) meeting these local conditions.

**Lemma 4.** *Assuming the true DAG $G$ adheres to the Erdös-Rényi graph model $G(d, p)$. Assuming that the prior knowledge $\pi$ leads to the weight deviation $\gamma$, which is sufficiently large, then the expected number of triangular patterns resulting from $\gamma$ approaches $\frac{2}{3}(d-2)p$, and the number of double collision patterns approaches $\frac{1}{6}(d-2)(d-3)p^2$.*

According to the lemma below, the number of both triangular and double collision patterns significantly surpasses their natural occurrence in the DAG:

**Lemma 5.** *In the ER graph $G(d, p)$, the expected number of natural triangular patterns and double collision patterns is $\frac{1}{6}d(d-1)(d-2)p^3$ and $\frac{1}{24}d(d-1)(d-2)(d-3)p^4$, respectively.*

The expected number of the two patterns given in Lemma 4 can also be approximately regarded as $\frac{2}{3}dp$ and $\frac{1}{6}(dp)^2$. This result indicates that the frequency of two graph patterns can be characterized by $dp$, which is numerically close to the average degree of the graph and characterizes the sparsity of the graph. Thus, we can further deduce that triangular patterns predominate when the average node degree is low, while double collision patterns become more frequent as the average degree increases.

### 4.3 Pattern-Guided Adaptive Prior Framework

Thus far, we have theoretically established how deviation $\gamma$ on a prior-enforced edge $(X_\beta \to X_j)$ can propagate (Lemma 2) to induce spurious edges $(X_k, X_j)$ when nodes $X_k, X_\beta$ satisfy certain structural relationships (Lemma 3), leading to characteristic triangle or double collision patterns. We also showed that these patterns strongly indicate deviation-induced errors. (Lemmas 4, 5). Leveraging these insights, this section introduces the Pattern-Guided Adaptive Prior framework, using these structural signals to mitigate adverse effects and foster more accurate weight learning.

The core idea of PGAP is to employ a two-stage process. First, we learn an initial DAG structure while incorporating prior knowledge. Second, we detect the number of deviation-induced patterns associated with each prior edge. If such patterns are found, we adaptively reduce the influence of the corresponding prior and re-optimize the model to obtain a more refined final graph. This procedure is detailed in Algorithm 1 for the common case where priors are integrated as soft constraints via a penalty term in the loss function.

---

**Algorithm 1** PGAP with Soft Prior Constraints

---

1: **Input:** Data matrix $X$, set of prior knowledge $\Pi$, initial prior loss weights $\alpha_0$ (vector with elements $\alpha_{0,\pi}$ for each $\pi \in \Pi$), pattern adjustment step size $\alpha$.
2: Learn $W_{\text{stage1}}$ using a base structure learning algorithm, incorporating the prior loss term $L_P(W; \Pi, \alpha_0)$.
3: Initialize $\alpha_1 \leftarrow \alpha_0$.
4: **for** each prior $\pi = (X_\beta \to X_j) \in \Pi$ **do**
5: $\quad n_{\text{patterns}} \leftarrow$ # of triangle and double collision patterns in $W_{\text{stage1}}$ involving $\pi$
6: $\quad \alpha_{1,\pi} \leftarrow \max(0, \alpha_{0,\pi} - \alpha \times n_{\text{patterns}})$
7: **end for**
8: Learn $W_{\text{final}}$ using the base structure learning algorithm, incorporating the prior loss term $L_P(W; \Pi, \alpha_1)$.
9: **Return** $W_{\text{final}}$

---

In this procedure, the objective function of the base learning algorithm is augmented with a prior loss term $L_P(W; \Pi, \alpha)$. Here, $\Pi$ denotes the set of provided priors, and $\alpha$ is a vector of weights governing the enforcement strength for each prior. The algorithm leverages the number of detected patterns, $n_{\text{patterns}}$, to adjust these weights, producing an updated weight vector $\alpha_1$ for the second optimization stage. This is discussed in more detail in Appendix C. This adaptive adjustment allows the model to counteract the effects of weight deviation by down-weighting priors that cause structural distortions. In addition, a similar implementation for hard prior constraints, which adjusts local regularization parameters, is also provided in Appendix C.

## 5 Experiments

This section presents a subset of the primary experimental results and analysis. We evaluate the performance of our proposed method across various synthetic data settings, including different graph

structures, numbers of nodes, dataset sizes, and noise types. We also assess its effectiveness when combined with different prior integration techniques and compare it against several baseline structure learning algorithms. Comprehensive details, extended analysis (including the impact of varying prior knowledge proportions and a detailed parameter analysis), supplementary experiments on real-world datasets, and explorations into non-linear data scenarios are available in Appendix D.

## 5.1 Experimental Setup

**Synthetic Data**   Random DAGs are generated using the Erdös-Rényi (ER) model or the Scale-Free (SF) model, with the number of nodes ($d$) set to 20, 40, and 60. We use ERk and SFk to denote ER and SF graphs, respectively, where $k$ represents the edge-to-node ratio. The edge-to-node ratio ($k$) varies between 2 and 4, corresponding to average degrees of 4 and 8, respectively. For graphs with 60 nodes, an edge-to-node ratio of $k = 8$ is also tested. The weighted adjacency matrix $B$ is created with uniform random weights drawn from $U(-2, -0.5) \cup U(0.5, 2)$, and the data is generated as $X = B^T X + \epsilon$. Noise $\epsilon$ is varied among Gaussian, exponential, and Gumbel distributions, referred to as Gauss, Exp, and Gumbel, respectively. Variance settings include equal variance (-EV) and non-equal variance (-NV). Sample sizes are set to 2 and 20 times the number of nodes (2d, 20d), and data is normalized post-generation. For the NV settings, the individual noise variances $\sigma_i^2$ for each variable $X_i$ were drawn independently from a uniform distribution $U(0.5, 2)$. All reported results on synthetic data represent the mean values obtained from six independent trials for each experimental configuration. All used priors are randomly selected from a proportion of edges in the true DAG.

**Methods**   We benchmark PGAP against several established structure learning methods, including VARSORT [Reisach et al., 2021], GOLEM [Ng et al., 2020], DAGMA [Bello et al., 2022], and NT-LogLL [Deng et al., 2024]. To demonstrate its adaptability and effectiveness, PGAP is integrated with each of these baseline methods, and its performance is compared against the standalone baseline. The least squares scoring function is used when noise is EV, and NLL is used when noise is NV. We also evaluate its performance when combined with different prior integration methods on NOTEARS. These include a hard prior integration method and two soft prior integration methods. The hard method specifies edge weights learned in the interval $(-\infty, -th) \cup (th, \infty)$ where $th$ is the threshold. The soft methods utilize two kinds of prior penalty functions:

$$\text{Softmax: } L_P(W; \Pi) = \sum W_\Pi - Softmax(|W| \odot W_\Pi)$$
$$\text{ReLU: } L_P(W; \Pi) = \sum Relu(th \times W_\Pi - |W| \odot W_\Pi) \tag{12}$$

In these equations, $W_\Pi$ represents the mask matrix derived from the prior knowledge, and $\sum$ denotes the summation over all elements of the matrix.

**Metrics**   F1 score, Structural Hamming Distance (SHD) or scaled SHD.

**Setup**   The initial weight of the prior loss is 1. The adaptive reduction parameter $\alpha$, i.e., the amount of weight reduction for each graph pattern, is 0.1. Other settings follow the default configurations of the baseline methods. Experiments are conducted on a system equipped with a 4.5 GHz AMD Ryzen 9 7950X CPU, NVIDIA GeForce RTX 3090 GPU, and 32GB of memory.

## 5.2 Experimental Results and Analysis

**Performance of PGAP in Various Graph Settings**   This experiment evaluates the performance of the PGAP by comparing it against the baseline NOTEARS-Softmax across a variety of graph settings. Figure 2 illustrates the performance comparison, showcasing results for different numbers of nodes, dataset sizes under different noise. Note, "scaled SHD" is calculated as the SHD divided by the number of nodes. Across these diverse conditions, the lines of proposed PGAP (the dark red and blue lines) are consistently positioned at the bottom of the plots and demonstrate superior performance. Results for denser graphs like ER4 and SF4 are detailed in Appendix D.1. The supplementary results also show that the improvement offered by PGAP tends to increase as the proportion of prior knowledge grows, contrasting with the baseline NOTEARS-Softmax, which can occasionally exhibit performance degradation with higher proportions of prior knowledge. PGAP helps to alleviate this issue, underscoring the importance of its mechanism for the reasonable integration of priors.

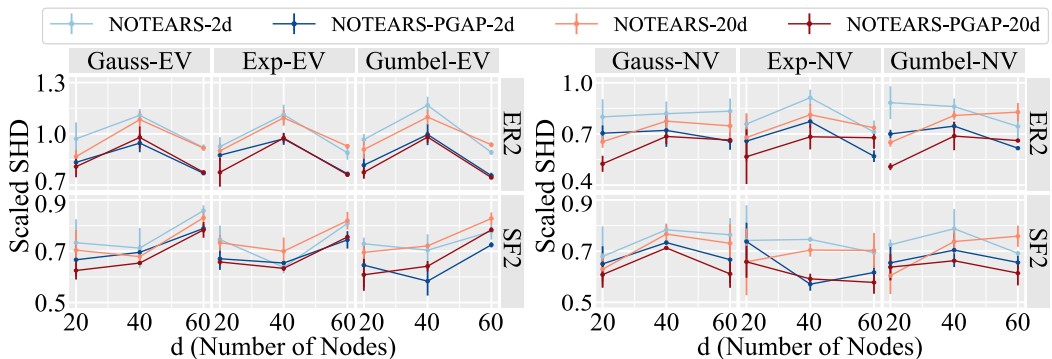

Figure 2: Performance of PGAP across Different Graph Settings (ER=2).

Table 1: Improvements of PGAP on different prior integration methods (Gauss-NV case).

| ER | Data Size | 2d | | | | 20d | | | |
|---|---|---|---|---|---|---|---|---|---|
| | Metric | F1 | | SHD | | F1 | | SHD | |
| | Prior Proportion | 0.2 | 0.4 | 0.2 | 0.4 | 0.2 | 0.4 | 0.2 | 0.4 |
| 2 | NOTEARS-Hard | 33.6±2.1 | 48.2±2.0 | 40.2±1.0 | 34.8±1.5 | 35.1±4.1 | 48.1±2.7 | 37.0±2.0 | 31.8±2.0 |
| | +PGAP | **37.0±3.0** | **53.9±2.1** | **37.0±2.1** | **29.3±1.4** | **35.4±2.5** | **52.1±4.4** | **35.3±1.7** | **28.3±2.3** |
| | NOTEARS-ReLU | 32.2±5.0 | 39.4±2.3 | 54.7±5.4 | 60.5±5.6 | 33.4±4.5 | 42.9±4.3 | 49.8±5.7 | 52.3±7.1 |
| | +PGAP | **34.9±3.7** | **42.3±3.5** | **49.8±4.1** | **53.0±8.0** | **35.1±4.7** | **47.0±4.5** | **46.8±4.9** | **42.0±9.4** |
| | NOTEARS-Softmax | 39.0±3.9 | 48.9±2.6 | 37.8±3.4 | 35.7±1.9 | 34.5±6.3 | 48.3±4.0 | 38.7±4.5 | 34.7±2.9 |
| | +PGAP | 37.7±2.9 | **57.8±5.4** | **35.2±1.5** | **26.7±2.3** | 33.4±2.2 | **56.4±5.1** | **37.5±1.8** | **26.3±2.9** |
| 4 | NOTEARS-Hard | 39.1±3.7 | 54.1±1.1 | 68.0±3.8 | 54.8±2.7 | 37.0±1.9 | 55.0±1.2 | 70.3±2.7 | 54.0±2.1 |
| | +PGAP | **39.8±3.7** | **55.5±2.0** | **66.2±4.3** | **52.2±3.5** | **37.6±2.9** | **55.5±2.4** | **69.3±3.3** | **53.2±2.7** |
| | NOTEARS-ReLU | 46.5±3.0 | 52.5±2.4 | 89.3±5.3 | 89.8±4.1 | 44.2±1.9 | 53.5±2.0 | 89.7±3.6 | 82.3±5.7 |
| | +PGAP | **46.9±3.0** | **53.2±2.2** | **86.3±5.0** | **86.8±4.9** | **45.4±2.3** | 53.3±1.4 | **86.3±4.2** | **80.3±5.6** |
| | NOTEARS-Softmax | 38.8±3.5 | 55.9±1.8 | 72.8±1.9 | 60.3±2.7 | 38.5±2.7 | 57.5±2.8 | 73.8±3.3 | 57.8±4.3 |
| | +PGAP | **40.4±4.0** | **56.9±1.5** | **69.0±2.9** | **57.0±2.4** | **39.0±1.6** | **60.4±2.9** | **69.0±2.8** | **50.8±4.6** |

**PGAP on Prior Integrating Methods** In this experiment, we integrate PGAP into 3 baseline prior integration methods: NOTEARS-Hard, -ReLU, and -Softmax, to evaluate the impact on their performance. The specific implementation of PGAP can be found in Appendix C. The results are detailed in Table 1. Additional performance data under various noise types are available in Appendix D.2. The results indicate that PGAP generally enhances the performance of these methods, particularly in terms of SHD, demonstrating its effectiveness across different prior integration methods.

**Comparison of PGAP with Established Methods** We integrated PGAP with several mainstream structure learning methods and compared these PGAP-enhanced versions against their original counterparts. The mainstream methods included in this comparison are NOTEARS, GOLEM, DAGMA, and NOTEARS-LogLL. For a fair comparison, these baseline methods, both in their original

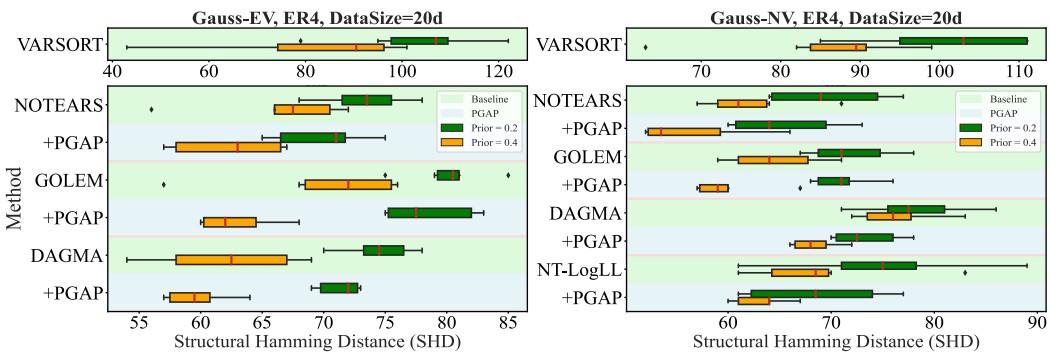

Figure 3: Improvements of PGAP on Mainstream Methods.

and PGAP-enhanced forms, utilized prior knowledge via a soft integration approach. Additionally, VARSORT is also included, which is a benchmark for structure learning. Comparative experiments were conducted across various prior proportions and noise types (see D.3), with the results presented in Figure 3. As shown in the figure, the PGAP-enhanced versions of the structural learning methods consistently outperform their respective original versions. This demonstrates PGAP's ability to improve upon existing methods by fostering a superior utilization of prior knowledge.

# 6   Limitation and Further Discussion

While our theoretical and empirical results demonstrate the effectiveness of the PGAP framework, it is important to acknowledge its limitations and the assumptions upon which it is built. A primary consideration is that our formal theoretical analysis is grounded in the linear SEM setting under the faithfulness assumption. Although our exploratory experiments suggest that PGAP remains beneficial in some nonlinear scenarios, extending the formal characterization of deviation propagation to general nonlinear models remains a challenging future direction.

Furthermore, the PGAP framework is designed as an adaptive, two-stage procedure. While it is theoretically motivated by the analysis of error propagation, the current work does not provide formal consistency guarantees for the resulting estimator. The framework's practical performance can also be context-dependent. As our analysis suggests, PGAP demonstrates robust improvements in sparse graph settings. However, in denser graphs where the target patterns may occur more frequently by chance, the framework's stability becomes a key consideration. We address this challenge and provide a more extensive discussion on it in Appendix D.8. Furthermore, a more extensive discussion of these aspects, alongside topics such as computational complexity and the handling of prior knowledge quality, is available in Appendix E.

# 7   Conclusion

This paper addresses the challenges of integrating prior knowledge into structure learning and offers practical solutions. Through theoretical analysis, we examine how edge weight deviation impacts the optimization process and specifically investigate two graph patterns whose prevalence increases with the magnitude of deviation. To counteract these issues, we introduce the Pattern-Guided Adaptive Prior (PGAP) Framework, designed to minimize the occurrence of these problematic patterns, thereby reducing the adverse effects of prior knowledge and improving its integration process. Our experimental results confirm the effectiveness and robustness of PGAP, demonstrating its superiority in DAG structure learning. This study highlights overlooked aspects of prior usage in existing research and pioneers the exploration of applying prior knowledge judiciously in continuous optimization.

## Acknowledgements

This research was supported in part by the National Key R&D Program of China (No. 2021ZD0111700), in part by the National Nature Science Foundation of China (No. 62137002, 62406302, 62176245), in part by the Natural Science Foundation of Anhui province (No. 2408085QF195), in part by the Fundamental Research Funds for the Central Universities under Grant WK2150110035.

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

# Appendix

## A  Extended Related Work

This section provides a more detailed discussion of Section 2 in the main paper. Subsection A.1 discusses the evolution of structure learning, while Subsection A.2 explores various methods and classifications aimed at enhancing the quality of structure learning through the integration of prior knowledge.

### A.1  The Development of Structure Learning

Structural learning seeks to delineate and comprehend the causal relationships between variables to enhance accuracy and reliability in prediction, decision-making, and intervention strategies. Numerous methods have been devised to unearth the true DAG structure from observational data. Constraint-based methods like PC [Spirtes et al., 2001] and FCI [Colombo et al., 2012] infer DAG structures through conditional independence tests between variables. Score-based methods evaluate the fit between a candidate DAG and the data using scoring functions. The hybrid methods, such as MMHC [Tsamardinos et al., 2006], merge both conditional independence tests and score-based methods. Additionally, some studies leverage data attributes such as non-Gaussianity, nonlinearity, and discreteness to investigate the DAG structure, as noted in several studies [Shimizu et al., 2006, Hoyer et al., 2008, Gretton et al., 2009, Peters et al., 2011, Ghoshal and Honorio, 2018, Khemakhem et al., 2021].

Recently, [Zheng et al., 2018] introduced NOTEARS, transforming the combinatorial search into a continuous optimization problem for the first time. This innovation streamlines the handling of acyclic constraints and the search of the vast DAG space, enabling the use of deep learning optimizers in the search process. Inspired by this breakthrough, subsequent research has expanded on continuous optimization methods for structure learning [Chen et al., 2025]. For instance, [Wei et al., 2020] expanded the algebraic characterization of DAG acyclicity to a broader class of matrix polynomials, while [Ng et al., 2019] developed GOLEM, which employs soft acyclicity constraints to enhance optimization performance. [Bello et al., 2022] introduced DAGMA, noted for its efficacy in detecting longer cycles and providing better gradients and faster performance.

Extensions of continuous optimization have also been adapted to diverse scenarios. For instance, [Zhu et al., 2020] and [Wang et al., 2021a] applied reinforcement learning to search for DAG structures, whereas [Yu et al., 2019] employed graph-based deep generative models to capture the variable distributions using the variational inference mechanism's evidence lower bound as an alternative to traditional scoring functions. [Lachapelle et al., 2019] utilized neural networks to model nonlinear relationships between variables, among other innovative approaches [Goudet et al., 2018, Kalainathan et al., 2022, Zheng et al., 2020].

Additionally, some studies have focused on refining the optimization mechanisms of NOTEARS. [Wei et al., 2022] explored the KKT optimality condition to better understand the behavior of NOTEARS and related algorithms. [Reisach et al., 2021] demonstrated that the performance of certain continuous structure learning methods might be attributed to the consistency between the increase in marginal variance and causal ordering. Continuing this line of inquiry, [Deng et al., 2023] introduced a likelihood-based scoring function that incorporates quasi-MCP regularization, offering scale-invariant characteristics.

### A.2  Structure Learning with Prior Knowledge

In practical applications, the scarcity of data and the presence of noise often hinder accurate results, prompting many studies to integrate human expertise or findings from previous research as prior knowledge into structure learning. There have been many works on structure learning under the combinatorial search framework.

Some studies enforce that the learned structure strictly adheres to prior knowledge, known as hard constraints. For instance, [De Campos and Castellano, 2007] adapted both the score-based hill climbing algorithm and the constraint-based PC algorithm to produce graphs consistent with hard constraints. Similarly, the Branch & Bound algorithm [De Campos et al., 2009] specifies mandatory and prohibited edges. More recently, some researchers have utilized ancestry as a hard constraint.

[Chen et al., 2016] combined the branch reduction capability offered by the prior with the A* search method to explore the application of hard constraints in precise searches. Related studies include [Borboudakis and Tsamardinos, 2012, Li and Beek, 2018, Wang et al., 2021b].

Alternatively, soft constraint methods use prior knowledge to guide the structure learning process. [Castelo and Siebes, 2000] assessed candidate DAGs based on prior knowledge concerning the presence or absence of edges between variables, as specified by experts. [Borboudakis and Tsamardinos, 2014] employed the joint prior probability distribution of ancestral relationships to score candidate DAGs. [Eggeling et al., 2019] conducted a broader evaluation of prior knowledge on general graph characteristics, such as restrictions on maximum in-degree and balancing different in-degree probabilities.

In the realm of continuous optimization, efforts to integrate prior knowledge continue, which can also be categorized into soft and hard constraint methods. Hard constraint methods focus on learning edge weights within a specified feasible range. For example, [Sun et al., 2023] utilized the L-BFGS-B optimization algorithm to set boundaries on weighted adjacency matrix parameters by establishing prior upper and lower bounds. [Chen and Ge, 2022] improved the NOTEARS framework using mixed integer linear models, incorporating prior knowledge through logical constraints and a series of equality/inequality constraints. Similar approaches were also adopted by [Ramsey et al., 2018] and [Liang et al., 2023].

Soft constraint methods augment the original objective with a new loss function measuring DAG-prior consistency and optimize continuously during the learning process. For instance, [Yang et al., 2023] constructed a loss function tied to a prior matrix that represents edge priors, ensuring that the weighted adjacency matrix, when masked by the prior matrix, closely resembles the prior matrix to include indicated edges. Similarly, [Wang et al., 2024] apply the ReLU function to the prior mask to optimize the degree to which edge weights fall below a fixed threshold, and this approach could be extended to softer activation functions. [Bello et al., 2022] manipulated the gradient of the objective function directly. They adjust the impact of individual elements on the sparse penalty term during updates based on prior knowledge to ensure specific edges are included in the result.

While these methods can generate DAGs qualitatively consistent with prior knowledge, they often overlook the nuanced learning of edge weights during optimization, which is a key distinction between continuous optimization and combinatorial search. Hard constraint methods exclude edge weights below a certain threshold to meet edge priors by setting feasible intervals, but they may fail to discern the positive or negative effects between variables, potentially leading to inappropriate weight adjustments during optimization. The effectiveness of soft constraint methods relying on prior loss varies with the loss function's stringency. Mild loss functions may struggle to shift the data's structural tendencies, making it challenging to satisfy prior knowledge. Conversely, when the loss function strongly emphasizes prior satisfaction, it may excessively increase corresponding edge weights, diverging from the true weights.

In summary, while these methods offer avenues to integrate causal knowledge into continuous objective functions, the resulting edge weights can significantly deviate from the ground truth. This discrepancy may have severe implications. To address this, our paper proposes a posterior strategy that examines the outcomes when the prior is either too strong or too weak, offering insights into how prior knowledge should guide learning.

## B   Proof of the Lemmas in the Main Text

The theoretical framework presented in this paper is grounded in the linear SEM, a widely adopted model for causal discovery. The validity and interpretation of our results rest on several key assumptions, which we delineate here for clarity.

**Assumptions for the Linear SEM (Eq. 1)**

Our model, defined as $X = B^T X + \epsilon$, is based on two primary assumptions. First is "Linearity," which posits that the causal relationship between a variable $X_i$ and its parents $Pa(X_i)$ is a linear combination, weighted by the elements of the true adjacency matrix $B$. While a simplification of many real world processes, this assumption provides crucial tractability and interpretability. Second is "Acyclicity," which requires the underlying graph structure to be a DAG. This is a cornerstone

assumption in most graphical causal models, ensuring that a variable cannot be its own ancestor and thus precluding feedback loops.

**Assumptions for the Scoring Functions (Eq. 3)**

The optimization objective for learning a candidate graph $W$ relies on a scoring function, such as $Q_{ls}(W; X)$ or $Q_{nll}(W; X)$. The choice between these scores implies assumptions about the noise distribution. The least squares score, $Q_{ls}$, is statistically equivalent to the (negative) log-likelihood under the assumption of independent and identically distributed Gaussian noise terms, i.e., $\epsilon_i \sim \mathcal{N}(0, \sigma^2)$ for all $i$. The more general negative log-likelihood score, $Q_{nll}$, relaxes this by allowing for differing variances across noise terms, i.e., $\epsilon_i \sim \mathcal{N}(0, \sigma_i^2)$.

Furthermore, we wish to emphasize that the core deviation propagation mechanism identified in our work appears robust even when these statistical assumptions are not perfectly met. As demonstrated in our experiments with non-Gaussian noise (e.g., Exponential, Gumbel), PGAP consistently provides improvements. This suggests that the identified triangular and double collision patterns are fundamental indicators of prior-induced error, and their utility is not strictly confined to scenarios where the model is perfectly specified. This robustness is a key strength of our proposed framework.

**Additional Notations and Formulations**

Before proving the lemmas, we first introduce notations not used in the main text. Given that $G$ is an acyclic graph, we can establish a topological order for the variables in $X$, ensuring no edges from lower to higher order variables. This ordering implies that $B$ can be represented as a strictly upper triangular matrix. According to Eq. (1), random variables $X$ can be expressed as a linear combination of noises $\epsilon$:

$$X = (I - B^T)^{-1}\epsilon \tag{13}$$

The matrix $I - B^T$ is invertible due to its upper triangular structure with non-zero diagonal elements. Let $\Sigma$ represent the covariance matrix of $X$, $\Sigma = \text{Cov}(X) = E[XX^T]$. It is related to $\Omega$ as follows:

$$\Sigma = (I - B)^{-T}\Omega(I - B)^{-1} \tag{14}$$

These equations elucidate the relationship between the causal structure $B$, the noise characteristics $\Omega$, and the resulting covariance $\Sigma$. The inverse of the covariance matrix, $\theta = \Sigma^{-1}$, is known as the precision matrix. Submatrices of $\Sigma$ and $\theta$ corresponding to specific subsets of variables play a key role in the derivation.

### B.1 The Proof of Lemma 1

**Lemma 1.** *Given a node $X_j$, assume that $t_j^*$ represents the set of predecessors of $X_j$ consistent with an order of the true DAG $G$. The regression coefficient obtained by linearly regressing $X_j$ on $t_j^*$ corresponds directly to the relevant element in the adjacency matrix $B$ of the true DAG, i.e.:*

$$B_{t_j^*,j} = \arg\min_{W_{t_j^*,j}} \left(\mathbf{X}_{t_j^*} W_{t_j^*,j} - \mathbf{X}_j\right)^2 \tag{15}$$

*Proof.* To establish the claim, we need to demonstrate that the optimal weight vector in the minimization problem, which corresponds to the OLS solution, is equal to $B_{t_j^*,j}$. The OLS solution is given by $(\Sigma_{t_j^*,t_j^*})^{-1}\Sigma_{t_j^*,j}$. We need to show:

$$B_{t_j^*,j} = (\Sigma_{t_j^*,t_j^*})^{-1}\Sigma_{t_j^*,j} \tag{16}$$

Given that $t_j^*$ is consistent with the order of the true DAG $G$, and utilizing the SEM property $\Sigma = (I - B)^{-T}\Omega(I - B)^{-1}$, we have:

$$\Sigma_{t_j^*,t_j^*} = (I - B_{t_j^*,t_j^*})^{-\top}\Omega_{t_j^*,t_j^*}(I - B_{t_j^*,t_j^*})^{-1} \tag{17}$$

Additionally, the covariance between $X_{t_j^*}$ and $X_j$ is given by $\Sigma_{t_j^*,j} = E[X_{t_j^*}X_j^\top]$, It can be further expressed as:

$$
\begin{aligned}
\Sigma_{t_j^*,j} &= E\left[X_{t_j^*}X_j^\top\right]\\
&= E\left[\left(I - B_{t_j^*,t_j^*}^\top\right)^{-1}\epsilon_{t_j^*}\epsilon^\top\left(I - B\right)^{-1}e_j\right]\\
&= \left(I - B_{t_j^*,t_j^*}\right)^{-\top}\left(\Omega_{t_j^*}\quad 0\right)\left(I - B\right)^{-1}e_j
\end{aligned}
\tag{18}
$$

where $e_j$ is the standard basis vector corresponding to node $X_j$. Thus, the regression coefficients can be obtained as follows:

$$
\begin{aligned}
&\left(\Sigma_{t_j^*,t_j^*}\right)^{-1}\Sigma_{t_j^*,j}\\
&= (I - B_{t_j^*,t_j^*})\Omega_{t_j^*}^{-1}(I - B_{t_j^*,t_j^*})^\top\left(I - B_{t_j^*,t_j^*}\right)^{-\top}\left(\Omega_{t_j^*}\quad 0\right)\left(I - B\right)^{-1}e_j\\
&= (I - B_{t_j^*,t_j^*})\Omega_{t_j^*}^{-1}\left(\Omega_{t_j^*}\quad 0\right)\left(I - B\right)^{-1}e_j\\
&= (I - B_{t_j^*,t_j^*})\left(I\quad 0\right)\left(I - B\right)^{-1}e_j\\
&= B_{t_j^*,j}
\end{aligned}
\tag{19}
$$

Hence, we complete the proof of Lemma 1. $\qquad\square$

## B.2 The Proof of Lemma 2

**Lemma 2.** *Let $\gamma = \hat{W}_{\beta,j} - W_{\beta,j}^*$ represent the deviation of the prior-influenced weight $\hat{W}_{\beta,j}$ (derived from (6)) from the optimal OLS weight $W_{\beta,j}^*$ obtained when regressing $X_j$ onto the set of predecessors $t_j$ (derived from (9)). The difference between the prior-influenced weights for other incoming edges $W_{t_j-\beta,j}^*$ and $\hat{W}_{t_j-\beta,j}$ is a linear function of $\gamma$, expressed as:*

$$
\hat{W}_{t_j-\beta,j} - W_{t_j-\beta,j}^* = \frac{\gamma}{\theta_{\beta,\beta}}\theta_{t_j-\beta,\beta}
\tag{20}
$$

*where $\theta$ is the inverse of the covariance matrix of the variable set $t_j$.*

*Proof.* According to the definitions of ordinary least squares (OLS) estimates in the main text, the optimal weights when regressing $X_j$ on $X_{t_j}$ are:

$$
W_{t_j,j}^* = \arg\min_{W_{t_j,j}}\left(\mathbf{X}_{t_j}W_{t_j,j} - \mathbf{X}_j\right)^2
\tag{21}
$$

The weights for $X_{t_j-\beta}$ when $W_{\beta,j}$ is fixed to $\hat{W}_{\beta,j}$ are denoted $\hat{W}_{t_j-\beta,j}$:

$$
\hat{W}_{t_j-\beta,j} = \arg\min_{W_{t_j-\beta,j}}\left(\mathbf{X}_{t_j-\beta}W_{t_j-\beta,j} + \mathbf{X}_\beta\hat{W}_{\beta,j} - \mathbf{X}_j\right)^2
\tag{22}
$$

The OLS solutions of Eq. (21) and (22) can be expressed using covariances:

$$
W_{t_j,j}^* = \left(\Sigma_{t_j,t_j}\right)^{-1}\Sigma_{t_j}^j
\tag{23}
$$

$$
\hat{W}_{t_j-\beta,j} = \left(\Sigma_{t_j-\beta,t_j-\beta}\right)^{-1}\mathrm{Cov}(X_{t_j-\beta}, X_j - \hat{W}_{\beta,j}X_\beta)
\tag{24}
$$

where

$$
\mathrm{Cov}(X_{t_j-\beta}, X_j - \hat{W}_{\beta,j}X_\beta) = \Sigma_{t_j-\beta,j} - \hat{W}_{\beta,j}\Sigma_{t_j-\beta,\beta}
\tag{25}
$$

So,

$$
\hat{W}_{t_j-\beta,j} = \left(\Sigma_{t_j-\beta,t_j-\beta}\right)^{-1}\left(\Sigma_{t_j-\beta,j} - \hat{W}_{\beta,j}\Sigma_{t_j-\beta,\beta}\right)
\tag{26}
$$

Let $W_{t_j-\beta,j}^*$ be the components of $W_{t_j,j}^*$ corresponding to $t_j - \beta$. We consider the case when $\hat{W}_{\beta,j}$ is set to the optimal OLS weight $W_{\beta,j}^*$ plus a deviation $\gamma$:

$$
\hat{W}_{\beta,j} = W_{\beta,j}^* + \gamma
\tag{27}
$$

The difference we want to calculate is $\hat{W}_{t_j-\beta,j} - W^*_{t_j-\beta,j}$.

$$\hat{W}_{t_j-\beta,j} - W^*_{t_j-\beta,j}$$
$$= \left(\Sigma_{t_j-\beta,t_j-\beta}\right)^{-1}\left(\Sigma^j_{t_j-\beta} - (W^*_{\beta,j}+\gamma)\Sigma_{t_j-\beta,\beta}\right) - W^*_{t_j-\beta,j} \tag{28}$$
$$= \left[\left(\Sigma_{t_j-\beta,t_j-\beta}\right)^{-1}\left(\Sigma^j_{t_j-\beta} - W^*_{\beta,j}\Sigma_{t_j-\beta,\beta}\right) - W^*_{t_j-\beta,j}\right] - \gamma\left(\Sigma_{t_j-\beta,t_j-\beta}\right)^{-1}\Sigma_{t_j-\beta,\beta}$$

The term in the square brackets is zero because it represents the difference when $\gamma = 0$, i.e., $\hat{W}_{\beta,j} = W^*_{\beta,j}$. In this case, $\hat{W}_{t_j-\beta,j}$ should equal $W^*_{t_j-\beta,j}$. (This assumes that regressing $X_j - W^*_{\beta,j}X_\beta$ onto $X_{t_j-\beta}$ gives the same coefficients $W^*_{t_j-\beta,j}$ as obtained from the joint regression on $X_{t_j}$. This holds due to properties of OLS.) Therefore,

$$\hat{W}_{t_j-\beta,j} - W^*_{t_j-\beta,j} = -\gamma\left(\Sigma_{t_j-\beta,t_j-\beta}\right)^{-1}\Sigma_{t_j-\beta,\beta} \tag{29}$$

The Eq. (29) shows that the difference is linear in $\gamma$. We proceed by calculating this difference in more detail. Define the following matrices for convenience:

$$M = \Sigma_{t_j,t_j},\ A_1 = \Sigma_{\beta,\beta},\ A_2 = \Sigma_{\beta,t_j-\beta},\ A_3 = \Sigma_{t_j-\beta,\beta},\ A_4 = \Sigma_{t_j-\beta,t_j-\beta} \tag{30}$$

Thus, the difference can be rewritten as:

$$\hat{W}_{t_j-\beta,j} - W^*_{t_j-\beta,j} = \gamma(A_4)^{-1}A_3 \tag{31}$$

In the above equation, the most difficult thing to deal with is the inverse of $A_4$. To this end, we first assume that $\beta = 1$ and then introduce the Schur complement, which provides us with:

$$M^{-1} = \begin{bmatrix} S^{-1} & -S^{-1}A_2A_4^{-1} \\ -A_4^{-1}A_3S^{-1} & A_4^{-1} + A_4^{-1}A_3S^{-1}A_2A_4^{-1} \end{bmatrix} \tag{32}$$

where $S$ is the Schur complement of $M$ with respect to $A_4$, defined as:

$$S = A_1 - A_2A_4^{-1}A_3 \tag{33}$$

From the conclusions of Eq. (32):

$$(M^{-1})_{2:j-1,2:j-1} = A_4^{-1} + A_4^{-1}A_3S^{-1}A_2A_4^{-1}$$
$$(M^{-1})_{2:j-1,1} = -A_4^{-1}A_3S^{-1} \tag{34}$$
$$(M^{-1})_{1,2:j-1} = -S^{-1}A_2A_4^{-1}$$

Using these conclusions, we can get the expression of the inverse of $A_4$:

$$A_4^{-1} = (M^{-1})_{2:j-1,2:j-1} - (M^{-1})_{2:j-1,1}(M^{-1})_{1,2:j-1} \cdot S$$
$$= (M^{-1})_{2:j-1,2:j-1} - (M^{-1})_{2:j-1,1}(M^{-1})_{1,2:j-1}((M^{-1})_{1,1})^{-1} \tag{35}$$

When $\beta \neq 1$, the permutation matrix $P$ can be used to transform $M$ to the case where $\beta = 1$:

$$M = P\Sigma_{t_j,t_j}P^T \tag{36}$$

Therefore the inverse of $M$ becomes:

$$M^{-1} = (P\Sigma_{t_j,t_j}P^\top)^{-1} = P(\Sigma_{t_j,t_j})^{-1}P^\top \tag{37}$$

The second equality in the above equation is due to the fact that the permutation matrix $P$ has the following property:

$$P^\top = P^{-1} \tag{38}$$

Therefore we can obtain:

$$A_4^{-1} = (M^{-1})_{2:j-1,2:j-1} - (M^{-1})_{2:j-1,1}(M^{-1})_{1,2:j-1}((M^{-1})_{1,1})^{-1}$$
$$= \theta_{t_j-\beta,t_j-\beta} - \theta_{t_j-\beta,\beta}\theta_{\beta,t_j-\beta}(\theta_{\beta,\beta})^{-1} \tag{39}$$

where

$$\theta = (\Sigma_{t_j,t_j})^{-1} \tag{40}$$

Therefore:
$$A_4^{-1}A_3 = \left(\theta_{t_j-\beta,t_j-\beta} - \theta_{t_j-\beta,\beta}\theta_{\beta,t_j-\beta}(\theta_{\beta,\beta})^{-1}\right)\Sigma_{t_j-\beta,\beta} \tag{41}$$

Next, we calculate the components separately:
$$\begin{aligned}
\text{First Term} &= e_{t_j-\beta}^\top \theta_{t_j,t_j-\beta}\Sigma_{t_j-\beta,t_j}e_\beta \\
&= e_{t_j-\beta}^\top\left(I - \theta_{t_j,\beta}\Sigma_{\beta,t_j}\right)e_\beta \\
&= -\theta_{t_j-\beta,\beta}\Sigma_{\beta,\beta}
\end{aligned} \tag{42}$$

$$\begin{aligned}
\text{Second Term} &= -\theta_{t_j-\beta,\beta}\theta_{\beta,t_j-\beta}(\theta_{\beta,\beta})^{-1}\Sigma_{t_j-\beta,\beta} \\
&= -\Sigma_{t_j-\beta,\beta}\theta_{\beta,t_j-\beta}\theta_{t_j-\beta,\beta}(\theta_{\beta,\beta})^{-1} \\
&= -\left(1 - \Sigma_{\beta,\beta}\theta_{\beta,\beta}\right)\theta_{t_j-\beta,\beta}(\theta_{\beta,\beta})^{-1} \\
&= -\theta_{t_j-\beta,\beta}(\theta_{\beta,\beta})^{-1} + \Sigma_{\beta,\beta}\theta_{t_j-\beta,\beta}
\end{aligned} \tag{43}$$

Consequently, by combining these two terms, we obtain:
$$A_4^{-1}A_3 = -\theta_{t_j-\beta,\beta}(\theta_{\beta,\beta})^{-1} \tag{44}$$

Finally, the difference in the estimates is expressed as:
$$\hat{W}_{t_j-\beta,j} - W_{t_j-\beta,j}^* = \gamma A_4^{-1}A_3 = \frac{\gamma}{\theta_{\beta,\beta}}\theta_{t_j-\beta,\beta} \tag{45}$$

This concludes the proof, showing that the difference between the adjusted and original weights is a linear function of the deviation $\gamma$, scaled by the term derived above. $\square$

## B.3  The Proof of Lemma 3

**Lemma 3.** *Assume $t_j = \{1, 2, \ldots, j-1\}$, and let $\theta$ be the inverse of the submatrix of the covariance of $X$ corresponding to the variable set $t_j$. Then the condition under which $\theta_{k,\beta}$ is non-zero is given by: $B_{k,\beta} \neq 0$ or $X_\beta$ and $X_k$ have a common child node $X_i$, when $k < \beta$; $B_{\beta,k} \neq 0$ or $X_\beta$ and $X_k$ have a common child node $X_i$, when $k > \beta$.*

*Proof.* Given that $t_j$ aligns with the true DAG order (say $t_j = \{1, 2, \ldots, j-1\}$), the inverse covariance matrix $\theta = (\Sigma_{t_j^*, t_j^*})^{-1}$ can be expressed in terms of the true adjacency matrix $B$ and noise variances $\Omega = \text{diag}(\sigma_1^2, \ldots, \sigma_d^2)$:

$$\theta = \left(I - B_{[j-1],[j-1]}\right)\Omega_{[j-1],[j-1]}^{-1}\left(I - B_{[j-1],[j-1]}\right)^\top \tag{46}$$

This leads to a specific formulation for the components of $\theta$:

$$\theta_{k,\beta} = \begin{cases} -\frac{B_{k,\beta}}{\sigma_\beta^2} + \sum_{i=\beta+1}^{j-1} \frac{B_{\beta,i}B_{k,i}}{\sigma_i^2}, & k < \beta \\ -\frac{B_{\beta,k}}{\sigma_k^2} + \sum_{i=k+1}^{j-1} \frac{B_{\beta,i}B_{k,i}}{\sigma_i^2}, & \beta < k < j \end{cases} \tag{47}$$

The analysis now splits based on the relative values of $k$ and $\beta$. Case 1: $k < \beta$. From Eq. (47), $\theta_{k,\beta}$ will be non-zero if: (1) The direct edge weight $B_{k,\beta}$ is non-zero. If $B_{k,\beta} \neq 0$, the first term $-\frac{B_{k,\beta}}{\sigma_\beta^2}$ contributes a non-zero value (assuming $\sigma_\beta^2 > 0$). (2) The common child term is non-zero. If $X_\beta$ and $X_k$ have a common child node $X_i$ such that $i \in \{\beta + 1, \ldots, j - 1\}$, then $B_{\beta,i} \neq 0$ and $B_{ki} \neq 0$. In this case, the summation term $\sum_{i=\beta+1}^{j-1} \frac{B_{\beta,i}B_{k,i}}{\sigma_i^2}$ is non-zero.

Case 2: $k > \beta$. From Eq. (47), $\theta_{k,\beta}$ will be non-zero if: (1) The direct edge weight $B_{\beta,k}$ is non-zero. If $B_{\beta,k} \neq 0$, the first term $-\frac{B_{\beta,k}}{\sigma_k^2}$ contributes a non-zero value. (2) The common child term is non-zero. If $X_\beta$ and $X_k$ have a common child node $X_i$ such that $i \in \{k + 1, \ldots, j - 1\}$, then $B_{\beta,i} \neq 0$ and $B_{ki} \neq 0$. The summation term $\sum_{i=k+1}^{j-1} \frac{B_{\beta,i}B_{k,i}}{\sigma_i^2}$ will be non-zero.

Combining these conditions yields the statement in the lemma. The exclusion of a Lebesgue zero-measure set corresponds to the faithfulness assumption, ensuring that terms don't accidentally cancel to zero. $\square$

## B.4 The Proof of Lemma 4

**Lemma 4.** *Assuming the true DAG $G$ adheres to the Erdös-Rényi graph model $G(d, p)$. Assuming that the prior knowledge $\pi$ leads to the weight deviation $\gamma$, which is sufficiently large, then the expected number of triangular patterns resulting from $\gamma$ approaches $\frac{2}{3}(d-2)p$, and the number of double collision patterns approaches $\frac{1}{6}(d-2)(d-3)p^2$.*

*Proof.* Assume that the prior knowledge producing the weight deviation is $(X_\beta \rightarrow X_j)$. To calculate the expected number of triangles generated by a given deviation $\gamma$, we denote this expectation as $E(\#\text{Triangle} \mid X_\beta \rightarrow X_j, \gamma)$. This expectation can be decomposed into expected sub-events:

$$E(\#\text{Triangle} \mid \pi = X_\beta \rightarrow X_j, \gamma) = \sum_{k<j} E\left(\text{Triangle}_{\beta k j} \mid X_\beta \rightarrow X_j, \gamma\right) \tag{48}$$

For cases where $k < \beta$:

$$E\left(\text{Triangle}_{\beta k j} \mid \pi = X_\beta \rightarrow X_j, \gamma\right) = P(X_k \rightarrow X_\beta)P(\gamma > \text{th}) \xrightarrow{\gamma \text{ is sufficiently large}} p \tag{49}$$

And for cases where $k > \beta$:

$$E\left(\text{Triangle}_{\beta k j} \mid \pi = X_\beta \rightarrow X_j, \gamma\right) = P(X_\beta \rightarrow X_k)P(\gamma > \text{th}) \xrightarrow{\gamma \text{ is sufficiently large}} p \tag{50}$$

Thus, for the prior knowledge $X_\beta \rightarrow X_j$, the expected number of triangles converges to:

$$E(\#\text{Triangle} \mid \pi = X_\beta \rightarrow X_j, \gamma) \rightarrow (j-2)p \tag{51}$$

To find the expectation over all possible prior knowledge $\pi$ in the graph, we compute the expectation:

$$E(\#\text{Triangle} \mid \pi, \gamma) = \frac{1}{C_d^2}\sum_{j=2}^{d}\sum_{\beta=1}^{j-1} E(\#\text{Triangle} \mid \pi = X_\beta \rightarrow X_j, \gamma) = \frac{2}{3}(d-2)p \tag{52}$$

Next, we address the expected number of double collision patterns, defined as:

$$E(\#\text{Double Collision} \mid \pi = X_\beta \rightarrow X_j, \gamma) \tag{53}$$
$$= \sum_{k<j}\sum_{k,\beta<i<j} E\left(\text{Double Collision}_{\beta k i j} \mid \pi = X_\beta \rightarrow X_j, \gamma\right)$$

For cases where $k < \beta$:

$$\sum_{k,\beta<i<j} E\left(\text{Double Collision}_{\beta k i j} \mid \pi = X_\beta \rightarrow X_j, \gamma\right) \tag{54}$$
$$= (j-1-\beta)P(X_\beta \rightarrow X_i)P(X_k \rightarrow X_i)P(\gamma > th) \xrightarrow{\gamma \text{ is sufficiently large}} (j-1-\beta)p^2$$

And for cases where $k > \beta$:

$$\sum_{k,\beta<i<j} E\left(\text{Double Collision}_{\beta k i j} \mid \pi = X_\beta \rightarrow X_j, \gamma\right) \tag{55}$$
$$= (j-1-k)P(X_\beta \rightarrow X_i)P(X_k \rightarrow X_i)P(\gamma > th) \xrightarrow{\gamma \text{ is sufficiently large}} (j-1-k)p^2$$

Summing the contributions for all $k$, the expected number of double collisions for prior knowledge $\pi = X_\beta \rightarrow X_j$ is:

$$E(\#\text{Double Collision} \mid \pi = X_\beta \rightarrow X_j, \gamma)$$
$$= \sum_{k=1}^{\beta-1}(j-1-\beta)p^2 + \sum_{k=\beta+1}^{j-1}(j-1-k)p^2 = \frac{1}{2}(j-\beta-1)(j+\beta-4)p^2 \tag{56}$$

Finally, we calculate the total expectation:

$$E(\#\text{Double Collision} \mid \pi, \gamma)$$
$$= \frac{1}{C_d^2}\sum_{\beta=1}^{d-1}\sum_{j=\beta+1}^{d} E(\#\text{Double Collision} \mid \pi = X_\beta \rightarrow X_j, \gamma) = \frac{1}{6}(d-2)(d-3)p^2 \tag{57}$$

Hence, we complete the proof of Lemma 4. $\qquad\square$

### B.5 The Proof of Lemma 5

**Lemma 5.** *In the ER graph $G(d, p)$, the expected number of natural triangular patterns and double collision patterns is $\frac{1}{6}d(d-1)(d-2)p^3$ and $\frac{1}{24}d(d-1)(d-2)(d-3)p^4$, respectively.*

*Proof.* We consider the probability that any three nodes in a DAG form a triangle pattern. Given the acyclic nature of DAGs, we assume a topological order for the nodes. Therefore, for three nodes a, b, c arranged in order in the Erdös-Rényi graph model $G(d, p)$, the probability that an edge exists between each pair of nodes is $p$, leading to a combined probability of $p^3$ for three interconnected nodes. Therefore, the expected number of triangle patterns in the DAG then calculated as:

$$E(\#\text{Triangle}) = C_d^3 \times p^3 = \frac{1}{6}d(d-1)(d-2)p^3 \tag{58}$$

Similarly, for any four nodes a, b, c, and d arranged in a DAG, the probability that nodes a and b simultaneously connect to nodes c and d in a structure resembling two collisions is $p^4$. Thus, the expected number of double collision patterns is:

$$E(\#\text{Double Collision}) = C_d^4 \times p^4 = \frac{1}{24}d(d-1)(d-2)(d-3)p^4 \tag{59}$$

$\square$

## C Algorithm for Pattern-Guided Adaptive Prior

This section details the implementation of the Pattern-Guided Adaptive Prior (PGAP) Framework. As a general strategy, PGAP integrates three key steps: (1) learning a DAG from data using a base structure learning algorithm, (2) detecting specific graph patterns that indicate potential prior-induced errors, and (3) adaptively adjusting parameters within the base learning algorithm to mitigate these errors. The adaptation method varies depending on how prior knowledge is incorporated.

### C.1 Implementation for Soft Prior Constraints

When prior knowledge is introduced as soft constraints via a prior loss term in the objective function, the adaptive adjustment in PGAP involves modifying the weight of this prior loss term. Algorithm 1 outlines the procedure.

---
**Algorithm 1** PGAP with Soft Prior Constraints

---
1: **Input:** Data matrix $X$, set of prior knowledge $\Pi$, initial prior loss weights $\alpha_0$ (vector with elements $\alpha_{0,\pi}$ for each $\pi \in \Pi$), pattern adjustment step size $\alpha$.
2: Learn $W_{\text{stage1}}$ using a base structure learning algorithm, incorporating the prior loss term $L_P(W; \Pi, \alpha_0)$.
3: Initialize $\alpha_1 \leftarrow \alpha_0$.
4: **for** each prior $\pi = (X_\beta \rightarrow X_j) \in \Pi$ **do**
5:     $n_{\text{patterns}} \leftarrow$ # of triangle and double collision patterns in $W_{\text{stage1}}$ involving $\pi$
6:     $\alpha_{1,\pi} \leftarrow \max(0, \alpha_{0,\pi} - \alpha \times n_{\text{patterns}})$
7: **end for**
8: Learn $W_{\text{final}}$ using the base structure learning algorithm, incorporating the prior loss term $L_P(W; \Pi, \alpha_1)$.
9: **Return** $W_{\text{final}}$

---

In this procedure, $L_P(W; \Pi, \alpha_i)$ represents the total prior loss, typically a sum of individual loss components for each prior:

$$L_P(W; \Pi, \alpha) = \sum_{\pi \in \Pi} \alpha_\pi L_P(W; \pi) \tag{60}$$

where $\alpha_\pi$ is the weight for prior $\pi$. Its specific form is problem-dependent, but it should satisfy:

$$L_P(W; \pi) \rightarrow 0 \Leftrightarrow \text{prior edge exists in the learned } G \tag{61}$$

The "base structure learning algorithm" refers to a chosen continuous optimization method (e.g., based on NOTEARS) that minimizes a scoring function $Q(W; X)$ subject to acyclicity $h(W) = 0$, augmented with the prior loss $L_P$ and a sparsity-inducing regularizer.

In this implementation, the weight $\alpha$ of the prior loss term $L_P$ is adaptively reduced when deviation-indicating patterns are detected. This lessens the influence of the prior, allowing the base structure learning algorithm more freedom to adjust edge weights based on the data.

### C.2 Implementation for Hard Prior Constraints

When prior knowledge is enforced using hard constraints (e.g., restricting edge weights to a specific range), the adaptive adjustment in PGAP involves modifying the local regularization strength. Algorithm 2 outlines this procedure.

---

**Algorithm 2** PGAP with Hard Prior Constraints

---

1: **Input:** Data matrix $X$, set of prior knowledge $\Pi$ (used as hard constraints), initial regularization matrix $\Lambda_0$ (with elements $\Lambda_0[k, j]$ e.g., 0.1), regularization adjustment step size $\omega$, maximum regularization strength $\lambda_{\max}$.
2: Learn $W_{\text{stage1}}$ using a base structure learning algorithm with hard prior constraints $\Pi$ and regularization matrix $\Lambda_0$.
3: Initialize $\Lambda_1 \leftarrow \Lambda_0$.
4: **for** each prior $\pi = (X_\beta \rightarrow X_j) \in \Pi$ **do**
5: $\quad$ $n_{\text{patterns}} \leftarrow$ # of triangle and double collision patterns in $W_{\text{stage1}}$ involving the $\pi$
6: $\quad$ **if** $n_{\text{patterns}} > 0$ **then**
7: $\quad\quad$ **for** each $k \neq j$ **do**
8: $\quad\quad\quad$ $\Lambda_1[k, j] \leftarrow \min(\Lambda_0[k, j] + \omega \times n_{\text{patterns}}, \lambda_{\max})$
9: $\quad\quad$ **end for**
10: $\quad$ **end if**
11: **end for**
12: Learn $W_{\text{final}}$ using the base structure learning algorithm with hard prior constraints $\Pi$ and the adjusted regularization matrix $\Lambda_1$.
13: **Return** $W_{\text{final}}$

---

Here, the regularization coefficient $\Lambda$ is adaptively increased for edges suspected of being spurious (i.e., participating in the detected patterns). This encourages these edge weights towards zero, promoting a sparser and more accurate graph.

In the experiment, each element of the regularization matrix $\Lambda$ starts at 0.1 with an upper limit of 0.2, and the regularization adjustment step size $\omega$ is 0.02.

## D Complete Experimental Results and Analysis

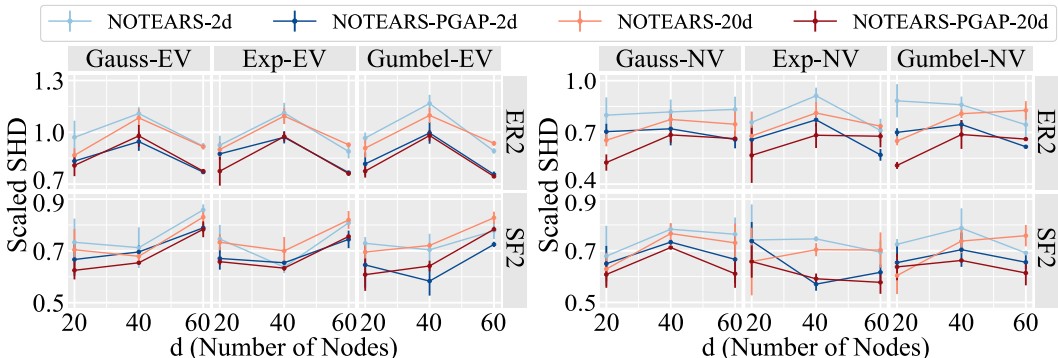

Figure 4: Performance of PGAP across Different Graph Settings (ER=2).

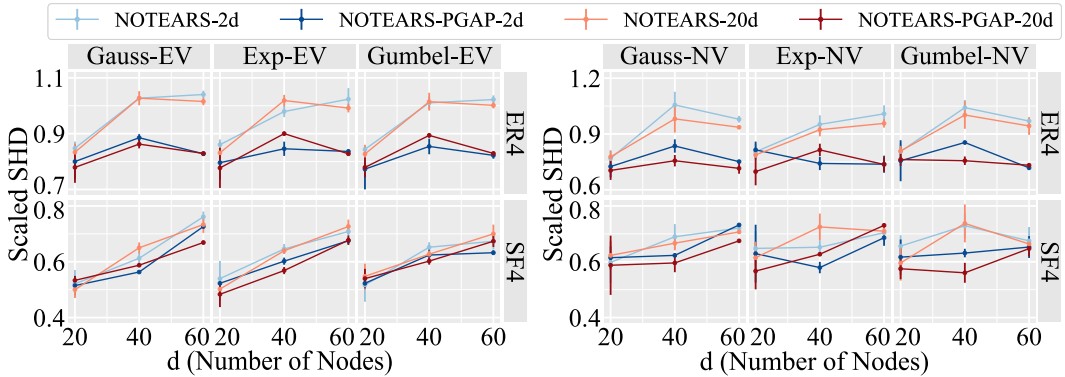

Figure 5: Performance of PGAP across Different Graph Settings (ER=4).

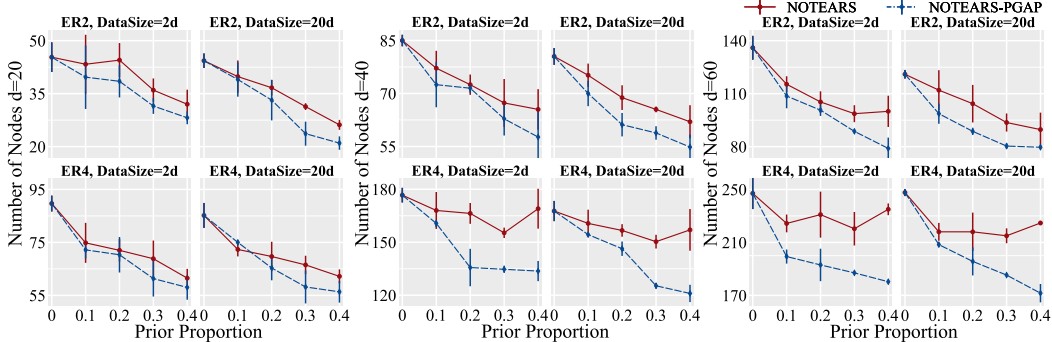

Figure 6: Performance Comparison across Different Prior Proportion Values (Gauss-NV case).

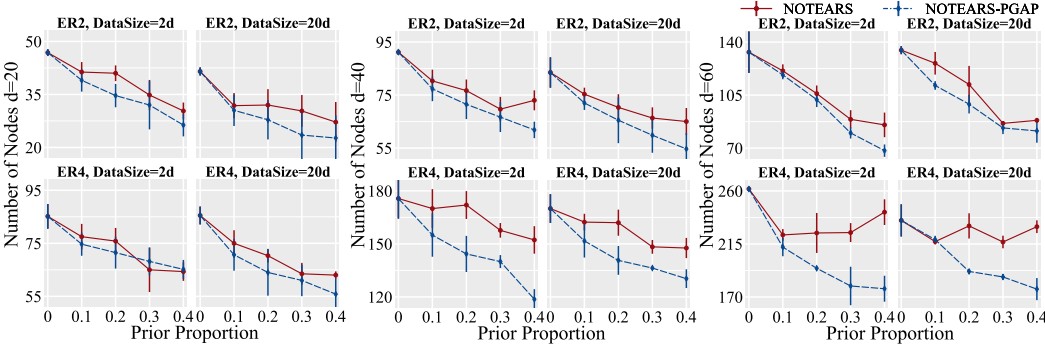

Figure 7: Performance Comparison across Different Prior Proportion Values (Exp-NV case).

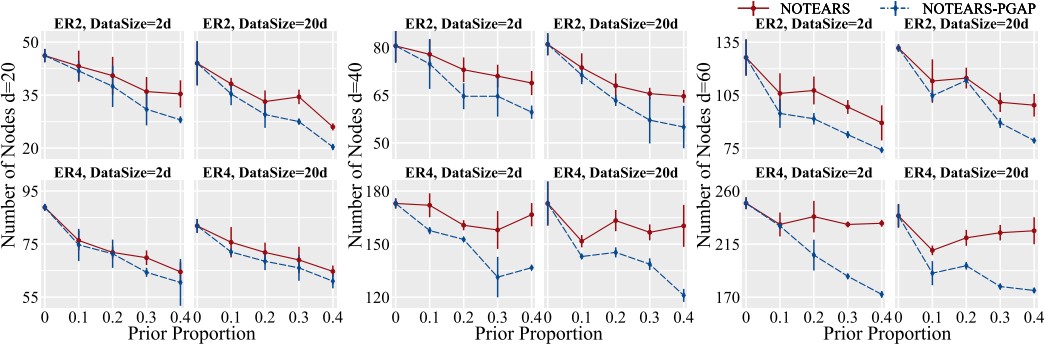

Figure 8: Performance Comparison across Different Prior Proportion Values (Gumbel-NV case).

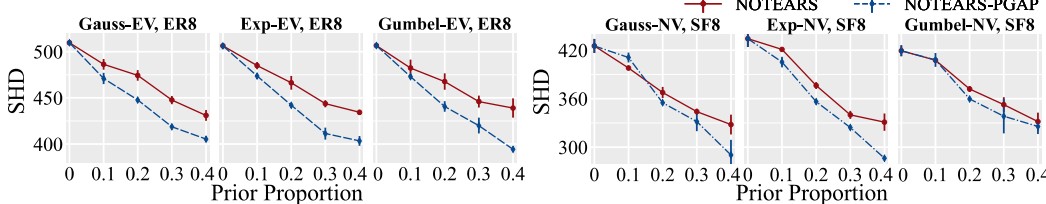

Figure 9: Performance Comparison on Large-scale denser Data (Nodes=60, ER=8).

## D.1 Performance of PGAP in Various Graph Settings

The main experiments (e.g., Figure 2 in the main text) have already established that PGAP demonstrates superior performance compared to the NOTEARS-Softmax baseline across a variety of settings for ER2 and SF2 graphs, including different numbers of nodes, dataset sizes, and multiple noise distributions (Gaussian-EV/NV, Exponential-EV/NV, Gumbel-EV/NV), using scaled SHD as the primary metric.

This appendix section provides further comprehensive details and extends this analysis to showcase PGAP's robustness and adaptability under additional conditions. The figures presented herein use scaled SHD, consistent with the main text. Specifically, this section expands on the findings by Presenting performance comparisons on denser graphs (ER4 and SF4), detailed in Figure 5. (Figure 4 reiterates the result of ER2 and SF2 scenarios for completeness). We also investigate the impact of varying proportions of prior knowledge on performance across different graph structures (ER graphs) and noise types (Gauss-NV, Exp-NV, Gumbel-NV), as illustrated in Figures 6, 7, and 8. Furthermore, we also evaluate the performance of PGAP on the denser graph with 60 nodes, as shown in Fig. 9.

These extended results (Figures 4 through 9) consistently show PGAP outperforming NOTEARS-Softmax. The enhancement in performance with an increasing proportion of prior knowledge further suggests that PGAP not only effectively utilizes prior knowledge but also adapts more gracefully to its integration, alleviating the occasional performance degradation observed with the NOTEARS-Softmax baseline. This highlights the critical role of PGAP's mechanism for judiciously handling prior knowledge to enhance rather than detract from the learning process.

## D.2 PGAP on Prior Integrating Methods

The experiment in the main section demonstrated that PGAP enhances the performance of NOTEARS-Hard, NOTEARS-ReLU, and NOTEARS-Softmax methods, with detailed results for the Gauss-NV noise case presented in Table 1.

This appendix further expands on these findings by providing performance data under additional non-equal variance (NV) noise conditions. Table 2 reiterates the Gauss-NV results for completeness within this appendix. The truly extended results, showcasing PGAP's impact with Exponential-NV (Exp-NV) and Gumbel-NV (Gumbel-NV) noise, are presented in Table 3 and Table 4, respectively.

These comprehensive results (Tables 2, 3, 4) are important for assessing the robustness and adaptability of PGAP in handling different noise structures when applied to various prior integration strategies. The findings indicate that PGAP generally maintains or improves both F1 scores and SHD relative to the baseline methods across most configurations. This reflects PGAP's capacity to effectively adapt to different noise characteristics while beneficially integrating prior knowledge into the structure learning process.

## D.3 Comparison of PGAP with Established Methods

The main text has discussed the comparison of the PGAP framework, when integrated with several established structural learning methods (NOTEARS, GOLEM, DAGMA, and NOTEARS-LogLL), against their original counterparts. Figure 3 in the main text provides a summary of these comparisons. This appendix section offers a more comprehensive and detailed graphical breakdown of these performance comparisons across a broader range of experimental conditions. The subsequent figures (Figures 10 through 15) illustrate these comparisons using SHD as the performance metric. These

Table 2: Improvements of PGAP on different prior integration methods (Gauss-NV case).

| ER | Data Size | 2d | | | | 20d | | | |
|---|---|---|---|---|---|---|---|---|---|
| | Metric | F1 | | SHD | | F1 | | SHD | |
| | Prior Proportion | 0.2 | 0.4 | 0.2 | 0.4 | 0.2 | 0.4 | 0.2 | 0.4 |
| 2 | NOTEARS-Hard | 33.6±2.1 | 48.2±2.0 | 40.2±1.0 | 34.8±1.5 | 35.1±4.1 | 48.1±2.7 | 37.0±2.0 | 31.8±2.0 |
| | +PGAP | **37.0±3.0** | **53.9±2.1** | **37.0±2.1** | **29.3±1.4** | **35.4±2.5** | **52.1±4.4** | **35.3±1.7** | **28.3±2.3** |
| | NOTEARS-ReLU | 32.2±5.0 | 39.4±2.3 | 54.7±5.4 | 60.5±5.6 | 33.4±4.5 | 42.9±4.3 | 49.8±5.7 | 52.3±7.1 |
| | +PGAP | **34.9±3.7** | **42.3±3.5** | **49.8±4.1** | **53.0±8.0** | **35.1±4.7** | **47.0±4.5** | **46.8±4.9** | **42.0±9.4** |
| | NOTEARS-Softmax | 39.0±3.9 | 48.9±2.6 | 37.8±3.4 | 35.7±1.9 | 34.5±6.3 | 48.3±4.0 | 38.7±4.5 | 34.7±2.9 |
| | +PGAP | 37.7±2.9 | **57.8±5.4** | **35.2±1.5** | **26.7±2.3** | 33.4±2.2 | **56.4±5.1** | **37.5±1.8** | **26.3±2.9** |
| 4 | NOTEARS-Hard | 39.1±3.7 | 54.1±1.1 | 68.0±3.8 | 54.8±2.7 | 37.0±1.9 | 55.0±1.2 | 70.3±2.7 | 54.0±2.1 |
| | +PGAP | **39.8±3.7** | **55.5±2.0** | **66.2±4.3** | **52.2±3.5** | **37.6±2.9** | **55.5±2.4** | **69.3±3.3** | **53.2±2.7** |
| | NOTEARS-ReLU | 46.5±3.0 | 52.5±2.4 | 89.3±5.3 | 89.8±4.1 | 44.2±1.9 | 53.5±2.0 | 89.7±3.6 | 82.3±5.7 |
| | +PGAP | **46.9±3.0** | **53.2±2.2** | **86.3±5.0** | **86.8±4.9** | **45.4±2.3** | 53.3±1.4 | **86.3±4.2** | **80.3±5.6** |
| | NOTEARS-Softmax | 38.8±3.5 | 55.9±1.8 | 72.8±1.9 | 60.3±2.7 | 38.5±2.7 | 57.5±2.8 | 73.8±3.3 | 57.8±4.3 |
| | +PGAP | **40.4±4.0** | **56.9±1.5** | **69.0±2.9** | **57.0±2.4** | **39.0±1.6** | **60.4±2.9** | **69.0±2.8** | **50.8±4.6** |

Table 3: Improvements of PGAP on different prior integration methods (Exp-NV case).

| ER | Data Size | 2d | | | | 20d | | | |
|---|---|---|---|---|---|---|---|---|---|
| | Metric | F1 | | SHD | | F1 | | SHD | |
| | Prior Proportion | 0.2 | 0.4 | 0.2 | 0.4 | 0.2 | 0.4 | 0.2 | 0.4 |
| 2 | NOTEARS-Hard | 40.6±7.6 | 49.5±1.6 | 35.3±5.3 | 32.2±1.5 | 35.8±2.4 | 51.7±4.0 | 37.7±1.2 | 31.0±1.8 |
| | +PGAP | 39.7±4.3 | **52.4±2.1** | **34.2±4.1** | **29.3±1.4** | 35.9±2.7 | **53.0±3.2** | **37.2±2.4** | **28.0±2.0** |
| | NOTEARS-ReLU | 32.6±3.0 | 41.6±4.0 | 54.7±5.8 | 57.8±8.7 | 33.8±3.1 | 43.0±3.1 | 51.5±2.9 | 54.3±3.1 |
| | +PGAP | 30.4±4.5 | **44.5±4.6** | **52.3±5.2** | **50.2±9.8** | **35.2±2.9** | **43.4±2.5** | **47.3±2.2** | 50.7±2.6 |
| | NOTEARS-Softmax | 33.2±7.0 | 47.2±6.5 | 41.7±5.6 | 36.7±5.7 | 33.3±3.6 | 48.4±4.3 | 40.8±2.6 | 35.8±3.3 |
| | +PGAP | **37.1±5.3** | **54.3±6.0** | **37.7±3.9** | **27.5±5.5** | **35.8±4.1** | **61.1±5.2** | **37.3±2.6** | **23.8±3.3** |
| 4 | NOTEARS-Hard | 39.5±1.5 | 55.1±3.1 | 67.3±2.7 | 53.3±4.0 | 38.2±3.1 | 54.5±3.0 | 69.8±3.3 | 55.2±2.6 |
| | +PGAP | **39.5±3.5** | **57.3±2.2** | 67.5±2.5 | **51.0±2.1** | **38.3±2.9** | **57.4±3.9** | **69.5±2.2** | **51.2±4.5** |
| | NOTEARS-ReLU | 46.7±3.0 | 54.7±2.8 | 84.7±5.7 | 78.2±7.0 | 46.2±3.3 | 53.8±1.7 | 88.0±7.3 | 83.3±5.1 |
| | +PGAP | **47.9±3.1** | **55.5±2.7** | **81.7±4.5** | **75.7±6.3** | **47.4±3.0** | **54.4±1.4** | **84.5±6.3** | **79.3±4.3** |
| | NOTEARS-Softmax | 41.6±4.8 | 57.7±1.6 | 69.7±6.9 | 58.2±2.6 | 37.7±1.8 | 57.0±1.0 | 74.8±3.3 | 59.2±3.0 |
| | +PGAP | 40.4±3.6 | **58.7±2.1** | **68.5±3.8** | **52.7±2.5** | **39.1±3.2** | **57.7±1.6** | **70.0±3.5** | **56.2±2.5** |

Table 4: Improvements of PGAP on different prior integration methods (Gumbel-NV case).

| ER | Data Size | 2d | | | | 20d | | | |
|---|---|---|---|---|---|---|---|---|---|
| | Metric | F1 | | SHD | | F1 | | SHD | |
| | Prior Proportion | 0.2 | 0.4 | 0.2 | 0.4 | 0.2 | 0.4 | 0.2 | 0.4 |
| 2 | NOTEARS-Hard | 36.9±3.5 | 47.6±4.0 | 38.0±1.4 | 33.3±2.9 | 38.9±6.9 | 47.1±2.5 | 36.3±5.1 | 33.2±2.1 |
| | +PGAP | **37.5±4.7** | **51.2±4.8** | **35.8±3.6** | **30.0±3.4** | 35.1±2.9 | **50.2±2.7** | 37.3±2.4 | **30.8±2.0** |
| | NOTEARS-ReLU | 31.4±4.2 | 42.9±3.3 | 55.7±6.3 | 55.3±6.2 | 33.5±4.2 | 42.4±3.5 | 52.0±3.9 | 54.5±4.3 |
| | +PGAP | **32.2±4.1** | **44.6±3.2** | **52.3±6.2** | **49.8±7.1** | **33.9±4.3** | **43.2±3.2** | **49.8±2.9** | **50.3±4.5** |
| | NOTEARS-Softmax | 32.6±4.6 | 48.7±3.1 | 41.8±2.5 | 37.3±3.3 | 34.2±4.9 | 49.7±7.0 | 40.5±3.6 | 36.3±5.5 |
| | +PGAP | 32.1±2.1 | **59.1±3.8** | **39.2±2.2** | **25.7±2.4** | **35.3±3.3** | **60.7±4.5** | **38.2±2.0** | **24.5±3.0** |
| 4 | NOTEARS-Hard | 40.0±1.9 | 53.4±2.8 | 68.0±2.0 | 56.7±3.4 | 38.6±4.5 | 54.5±2.7 | 69.7±3.1 | 55.0±2.8 |
| | +PGAP | 38.5±3.8 | **54.8±2.8** | 68.8±4.6 | **54.7±2.5** | 36.3±1.8 | **55.7±1.8** | 70.2±2.9 | **52.2±2.3** |
| | NOTEARS-ReLU | 44.9±2.9 | 53.5±3.8 | 89.8±3.8 | 86.7±8.1 | 43.9±2.2 | 53.2±2.2 | 92.7±6.0 | 85.2±3.2 |
| | +PGAP | **45.0±2.4** | **55.5±3.2** | **88.3±2.7** | **81.7±7.1** | **45.3±2.3** | **54.1±2.3** | **88.0±6.5** | **82.2±4.2** |
| | NOTEARS-Softmax | 37.2±4.2 | 56.4±2.2 | 74.3±3.6 | 59.3±3.7 | 37.2±4.2 | 57.5±1.7 | 75.0±6.2 | 57.0±2.7 |
| | +PGAP | **37.3±3.2** | **57.5±1.2** | **70.2±3.0** | **55.5±1.7** | **38.8±3.5** | **58.3±1.9** | **68.5±5.8** | **53.8±2.2** |

figures cover various settings, including different noise distributions with Equal Variance (EV) and Non-Equal Variance.

These detailed results visually reinforce the findings from the main text, demonstrating that the PGAP-enhanced versions of the structural learning methods consistently tend to outperform their respective original versions across these diverse scenarios, highlighting PGAP's ability to foster a superior utilization of prior knowledge.

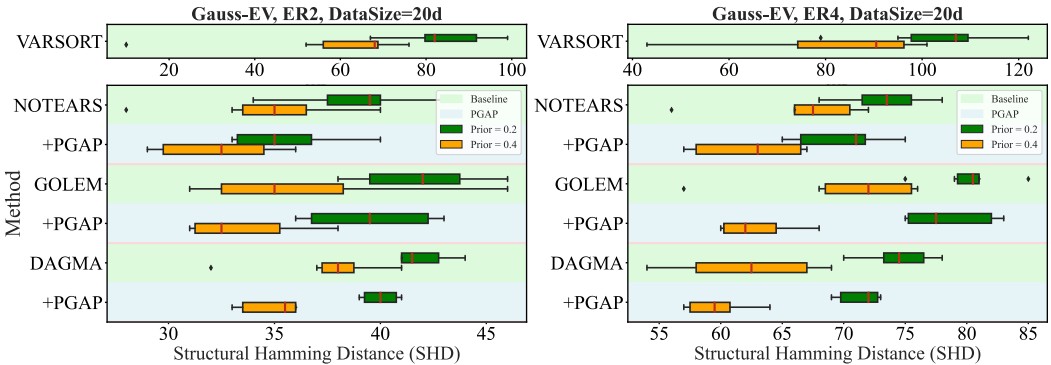

Figure 10: SHD Performance Comparison of PGAP with Mainstream Methods (Exp-EV case).

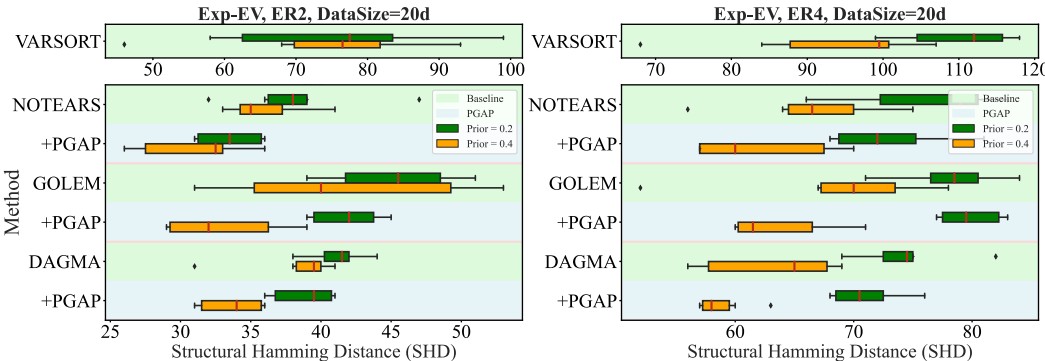

Figure 11: SHD Performance Comparison of PGAP with Mainstream Methods (Gauss-EV case).

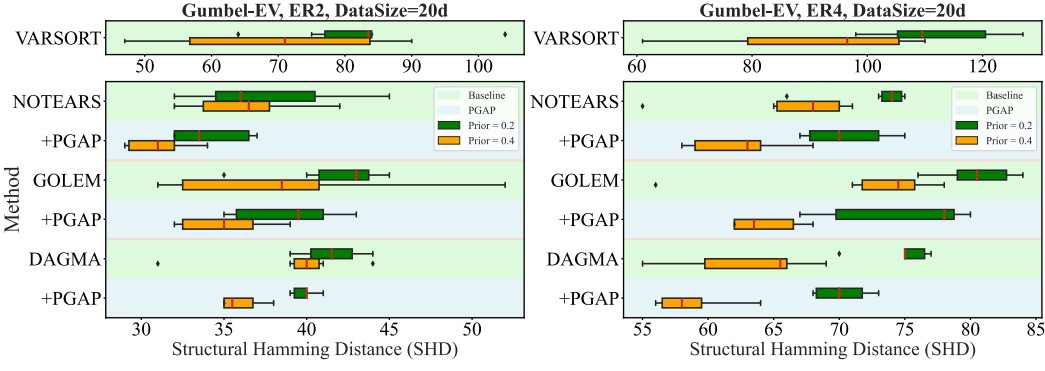

Figure 12: SHD Performance Comparison of PGAP with Mainstream Methods (Gumbel-EV case).

## D.4   Ablation Study on the Adaptive Reduction Parameter

The PGAP framework, when applied with soft prior constraints (as detailed in Algorithm 1), utilizes an 'adaptive reduction parameter', denoted as $\alpha$. This parameter determines the magnitude by which the weight of a specific prior in the loss function is reduced when deviation-indicating structural patterns

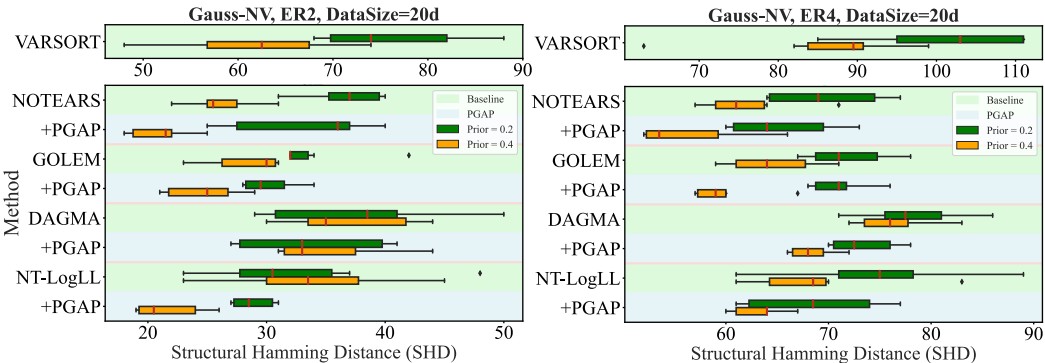

Figure 13: SHD Performance Comparison of PGAP with Mainstream Methods (Exp-NV case).

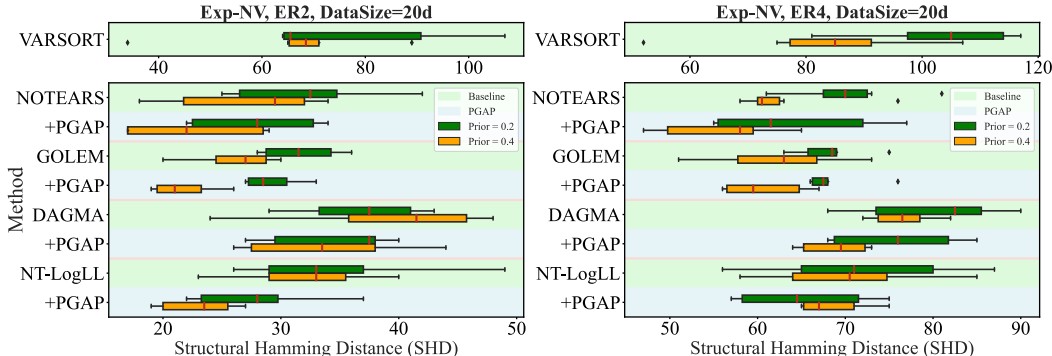

Figure 14: SHD Performance Comparison of PGAP with Mainstream Methods (Gauss-NV case).

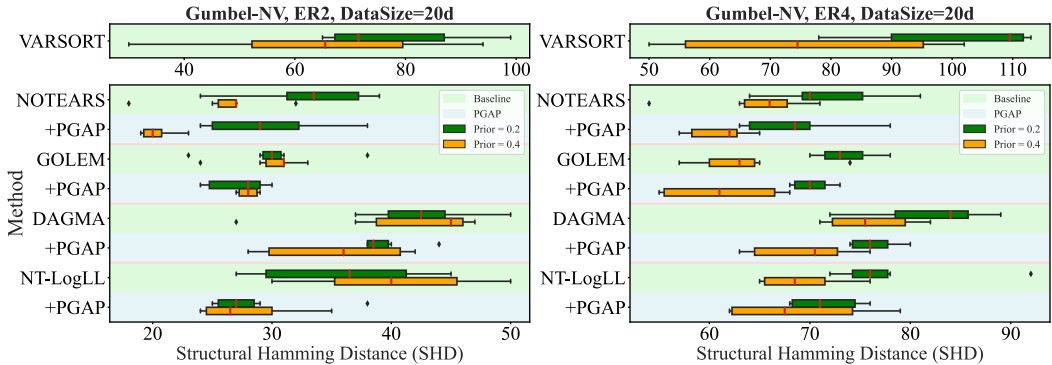

Figure 15: SHD Performance Comparison of PGAP with Mainstream Methods (Gumbel-NV case).

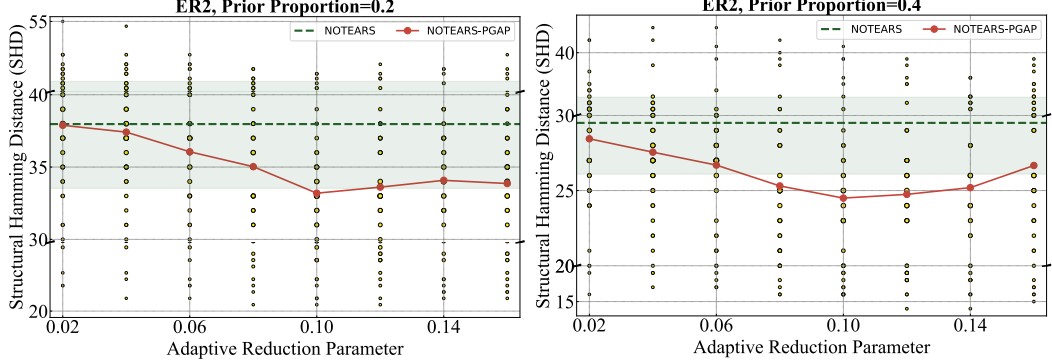

Figure 16: The Ablation Results for the Adaptive Reduction Parameter $\alpha$ (ER2).

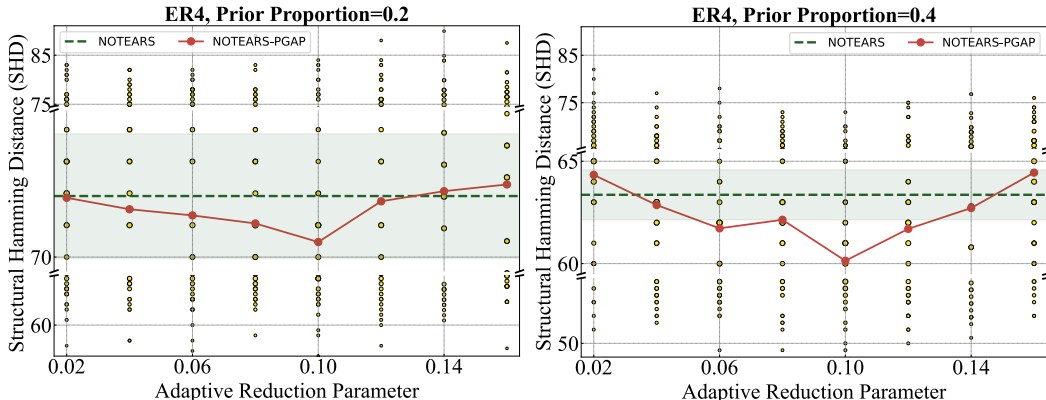

Figure 17: The Ablation Results for the Adaptive Reduction Parameter $\alpha$ (ER4).

associated with that prior are detected during the initial learning stage ($W_{stage1}$). Understanding the sensitivity of PGAP's performance to this $\alpha$ parameter is important for its practical application.

This ablation study investigates the impact of varying the adaptive reduction parameter $\alpha$ on the performance of PGAP, specifically when integrated with NOTEARS using a softmax-based soft prior (referred to as NOTEARS-PGAP). Figures 16 and 17 display the SHD for ER2 and ER4 graphs, respectively, under different prior proportions (0.2 and 0.4). In these figures, the numerous black scatter points represent the SHD values from individual runs of the NOTEARS with PGAP's adaptive mechanism.

The results show that NOTEARS-PGAP (solid red line) consistently achieves a lower average SHD than NOTEARS results across a range of $\alpha$ values. For instance, an adaptive reduction parameter $\alpha$ around 0.1 (the default value used in our other experiments) demonstrates robust performance, typically falling below the majority of the baseline scatter points. This indicates that PGAP's adaptive adjustment of prior loss weights provides a tangible benefit over a static prior weighting. While extreme values of $\alpha$ might lead to diminishing returns or slight increases in SHD, the mechanism shows a favorable range of operation.

This study supports the conclusion that PGAP's adaptive approach to modulating prior influence based on structural patterns is effective and not overly sensitive to the precise setting of the adaptive reduction parameter within a reasonable range, generally outperforming the non-adaptive baseline.

### D.5   Results on Real-World Data

To evaluate the practical applicability of our framework, we conducted experiments on four real-world datasets of varying scales and domains. The first is the well-established Sachs dataset, which contains 11 nodes and 17 edges from measurements of protein and phospholipid expression levels [Sachs et al., 2005]. We also include three additional datasets: the small-scale PaidSearch dataset (7 nodes, 6 edges) Rutz and Bucklin [2007], and two larger-scale datasets, Magic-niab (44 nodes, 66 edges) Scutari et al. [2014] and Ecoli70 (46 nodes, 70 edges) Schäfer and Strimmer [2005].

Table 5 presents the performance of three baseline methods (NOTEARS-Hard, -ReLU, and -Softmax) with and without PGAP enhancement on these four datasets, using F1 score and SHD as metrics with prior proportions of 0.2 and 0.4. The results demonstrate a consistent trend across all evaluated scenarios. Integrating PGAP generally leads to improved outcomes, reflected in higher F1 scores and lower SHD values. This robust performance across diverse real-world data underscores the practical value and generalizability of our proposed framework.

### D.6   Experiments on Nonlinear Situation

To further test the applicability of the proposed method in different scenarios, we also conducted experiments under nonlinear conditions, with the results presented in Table 6.

Table 5: Performance comparison on four real-world datasets: Sachs, PaidSearch, Magic-niab, and Ecoli70.

| Dataset | Sachs | | | | PaidSearch | | | | Magic-niab | | | | Ecoli70 | | | |
|---|---|---|---|---|---|---|---|---|---|---|---|---|---|---|---|---|
| Metric | F1 | | SHD | | F1 | | SHD | | F1 | | SHD | | F1 | | SHD | |
| Prior Proportion | 0.2 | 0.4 | 0.2 | 0.4 | 0.2 | 0.4 | 0.2 | 0.4 | 0.2 | 0.4 | 0.2 | 0.4 | 0.2 | 0.4 | 0.2 | 0.4 |
| NOTEARS-Hard | 48.0 | 53.3 | 11 | 10 | 15.4 | 42.9 | 9 | 7 | 34.9 | 46.7 | 51 | 45 | 26.7 | 35.0 | 123 | 112 |
| +PGAP | 48.0 | 57.1 | 11 | 9 | 15.4 | 46.2 | 9 | 6 | 35.7 | 54.2 | 50 | 40 | 28.4 | 38.7 | 107 | 103 |
| NOTEARS-ReLU | 48.0 | 55.2 | 11 | 10 | 13.3 | 30.8 | 11 | 10 | 35.3 | 42.7 | 51 | 47 | 23.0 | 29.5 | 161 | 176 |
| +PGAP | 48.0 | 57.1 | 11 | 9 | 16.7 | 42.9 | 8 | 7 | 37.2 | 53.2 | 49 | 41 | 27.9 | 35.7 | 117 | 99 |
| NOTEARS-Softmax | 37.0 | 48.5 | 13 | 13 | 30.8 | 40.0 | 8 | 8 | 35.3 | 44.9 | 51 | 46 | 20.6 | 29.0 | 155 | 176 |
| +PGAP | 40.0 | 51.9 | 12 | 10 | 33.3 | 46.2 | 7 | 6 | 37.2 | 54.7 | 49 | 40 | 25.2 | 36.0 | 113 | 108 |

Table 6: Improved Effect of PGAP on the Nonlinear Conditions (20 Nodes).

| Nonlinear Function Type | | MIM | | | | MLP | | | |
|---|---|---|---|---|---|---|---|---|---|
| Prior Proportion | | 0.2 | | 0.4 | | 0.2 | | 0.4 | |
| Graph Type | Method | F1 | SHD | F1 | SHD | F1 | SHD | F1 | SHD |
| ER2 | NOTEARS-MLP | 88.99 | 6.50 | 95.12 | 2.83 | 70.80 | 19.33 | 79.73 | 13.67 |
| | +PGAP | **89.75** | **5.83** | **96.21** | **2.17** | **73.05** | **17.33** | **82.54** | **11.83** |
| SF2 | NOTEARS-MLP | **68.45** | **16.33** | 76.10 | 12.50 | 53.19 | 25.50 | 66.62 | 19.50 |
| | +PGAP | 67.80 | 16.50 | **80.11** | **11.33** | **53.26** | **24.33** | **71.09** | **16.67** |
| ER4 | NOTEARS-MLP | 81.46 | 23.83 | 87.56 | 16.83 | 64.90 | 40.33 | 79.64 | 25.83 |
| | +PGAP | **82.81** | **22.50** | **88.42** | **15.67** | **65.95** | **38.50** | **81.01** | **23.83** |
| SF4 | NOTEARS-MLP | 66.16 | 32.83 | 82.22 | 19.67 | 48.78 | 50.00 | 65.81 | 36.83 |
| | +PGAP | **72.84** | **28.50** | **83.71** | **18.17** | **50.86** | **48.67** | **66.40** | **36.00** |

Compared to previous experiments, we employed nonlinear methods to generate data, including the MLP and MIM approaches. The MLP method applies the sigmoid function to introduce nonlinear transformations to the input $X$, while the MIM method uses trigonometric functions for nonlinear transformations. Specifically, the former can be expressed as $X \leftarrow \text{sigmoid}(XW_1)W_2 + z$, and the latter as $X \leftarrow \tanh(XW_1) + \cos(XW_2) + \sin(XW_3) + z$, where $W_1, W_2, W_3$ are randomly generated weight matrices, and $z$ is a random noise.

Table 6 compares the performance of the proposed method and NOTEARS-MLP under nonlinear conditions across different settings. The results show that even in non-linear cases, the proposed method still achieves effective performance improvements in most scenarios. However, we also observe that in some cases, the performance gains are not significant, which may be attributed to the complexity of variable dependencies in nonlinear settings, limiting the capability of the proposed method.

### D.7 Sample Analysis of Comparative Experiments

To illustrate the improvement offered by the PGAP method, we analyze a DAG with 10 nodes, ER=2, and Gauss-NV noise. Figures 18 showcase the DAGs learned using NOTEARS-Softmax and PGAP, called DAG-Softmax and DAG-PGAP. The discrepancies between the learned DAGs and the true DAG are indicated by different colored edges: black lines represent correctly inferred edges, red lines indicate extra edges not present in the true DAG. Dotted lines represent the incorporation of prior knowledge in the learning process.

DAG-Softmax results in 6 extra edges, while DAG-PGAP has fewer, with 3 extra edges, demonstrating PGAP's superior performance.

DAG-Softmax exhibits 3 triangles that include both prior knowledge and extra edges that point to the same node, including (5,2,1), (5,2,6), and (6,8,5). In contrast, DAG-PGAP shows a reduction in such triangle patterns, featuring 2 such instances: (5,2,1),(6,8,5). The structure (5,2,6) is identified, leading to the extra edge from 6 to 2 being deleted. These 2 triangles in DAG-PGAP involve just 2 extra edges. This reduction indicates that under the influence of PGAP, even when the condition of Lemma 3 is met, forming triangle patterns becomes more difficult.

Moreover, regarding the double collision pattern, DAG-Softmax displayed one instance: (6,8,3,9). In contrast, DAG-PGAP recorded no such instance. This comparison illustrates that PGAP effectively minimizes the number of special graph patterns, reinforcing its utility in refining the structure learning process by integrating prior knowledge more efficiently and reducing errors due to extra edges.

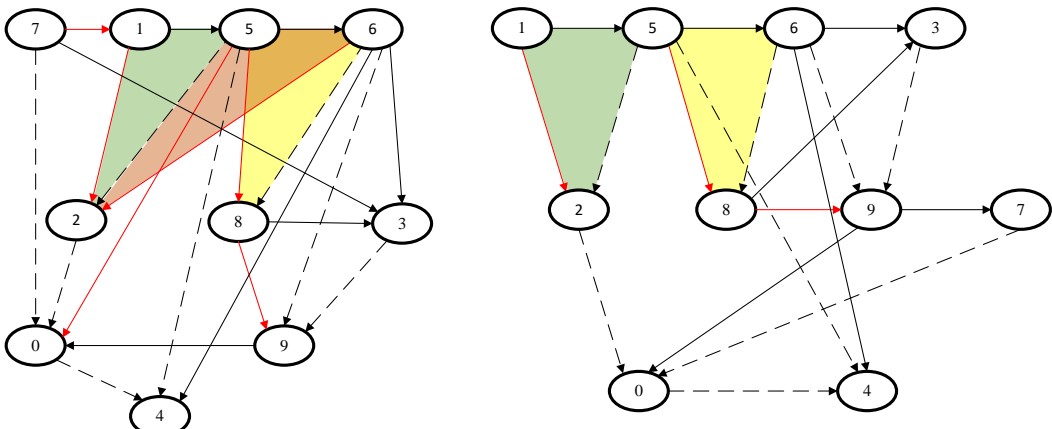

Figure 18: The structure learning results of NOTEARS-Softmax (left) and DAG-PGAP (right) for a DAG with 10 nodes, ER=2, and Gauss-NV noise. 40% priors are utilized.

### D.8 Performance on Dense Graphs with a Data-driven Adjustment Mechanism

As discussed in Section 6, the performance of the baseline PGAP adjustment can be sensitive in denser graphs. This is because the raw count of triangular and double collision patterns, used as the primary signal, may not be as informative when such patterns occur more frequently by chance in a highly connected structure. To address this challenge and enhance the robustness of PGAP, we introduce and validate a refined, data-driven adjustment mechanism that relies on a non-oracle tuning procedure.

This refined procedure normalizes the pattern count for a prior-enforced edge against a baseline derived empirically from the graph's own structure. The process is as follows. First, after learning an initial graph $W_{\text{stage1}}$, we iterate through every edge in the graph to compute its associated pattern count, which yields an empirical distribution of these counts for the specific graph. From this distribution, we establish a baseline for the "normal" level of patterns, $n_{\text{baseline}}$, using a robust statistic such as the median or the 75th percentile. For each prior-enforced edge $\pi$, we then calculate its excess pattern count: $n_{\text{excess}} = \max(0, n_{\text{patterns},\pi} - n_{\text{baseline}})$. This excess count, representing an anomalous number of patterns, is then used to determine the adjustment to the prior's weight $\alpha_\pi$. This non-oracle method ensures that the adjustment is sensitive only to pattern counts that are unusually high for the given graph's density, thereby stabilizing performance.

To validate this data-driven procedure, we conducted extensive experiments on dense graphs. Table 7 shows the SHD results for graphs with 20 and 40 nodes with high average degrees. Table 8 presents results for even more challenging scenarios with 60 nodes and an average degree of 24. We compare the baseline NOTEARS-Softmax against PGAP enhanced with our data-driven adjustment, using both the median and the 75th percentile as the baseline statistic.

The results consistently demonstrate that both data-driven variants of PGAP significantly outperform the baseline method across all tested dense graph scenarios and noise types. This confirms that the refined adjustment mechanism is not only theoretically sound but also a practical and effective solution for robustly applying PGAP in challenging, highly-connected environments.

### D.9 Performance on Diverse Graph Topologies

To further assess the generalizability of PGAP and ensure its effectiveness is not limited to ER or SF graphs, we conducted additional experiments on three distinct random graph models that generate DAGs with diverse structural properties. The Watts-Strogatz (WS) model produces graphs with high

Table 7: Performance of the data-driven PGAP adjustment on dense graphs (d=20, 40). SHD is reported.

| Noise | Node Graph&Edge | d=20 | | | | d=40 | | | |
| | | ER&4d | | SF&4d | | ER&8d | | SF&8d | |
| | Prior Proportion | 0.2 | 0.4 | 0.2 | 0.4 | 0.2 | 0.4 | 0.2 | 0.4 |
|---|---|---|---|---|---|---|---|---|---|
| Exp | NOTEARS-Softmax | 69.2±8.4 | 64.0±6.3 | 54.5±7.7 | 49.2±6.9 | 296.0±7.8 | 259.0±11 | 218.3±0.8 | 180.0±13 |
| | +PGAP(median) | 62.3±7.9 | 56.7±6.0 | 51.5±8.5 | 42.3±7.3 | 282.0±4.7 | 233.8±7.9 | 213.5±4.6 | 162.8±10 |
| | +PGAP(75th) | 62.2±7.6 | 59.0±6.0 | 52.2±8.2 | 43.2±7.0 | 282.7±5.2 | 231.2±8.3 | 209.2±5.4 | 159.7±11 |
| Gauss | NOTEARS-Softmax | 67.8±6.2 | 60.8±4.5 | 53.5±9.2 | 47.3±5.9 | 298.2±7.1 | 258.5±8.0 | 229.5±8.2 | 179.8±8.3 |
| | +PGAP(median) | 67.5±4.8 | 57.8±6.7 | 47.7±7.8 | 43.3±8.1 | 284.8±4.5 | 238.7±6.8 | 218.5±2.9 | 165.2±9.0 |
| | +PGAP(75th) | 66.7±4.5 | 55.7±8.3 | 48.8±9.5 | 43.2±7.0 | 286.8±4.3 | 238.5±8.6 | 214.0±6.1 | 159.2±11 |
| Gumbel | NOTEARS-Softmax | 71.8±6.1 | 63.3±5.3 | 53.2±8.9 | 47.8±9.4 | 294.5±6.3 | 256.8±7.6 | 222.0±5.9 | 181.5±10 |
| | +PGAP(median) | 66.3±6.7 | 59.0±6.7 | 47.8±8.5 | 40.0±8.5 | 280.5±6.5 | 238.7±5.9 | 219.0±1.8 | 164.2±9.2 |
| | +PGAP(75th) | 64.5±5.5 | 57.7±4.7 | 48.7±7.5 | 40.7±8.3 | 286.8±2.1 | 234.5±8.0 | 217.8±2.8 | 161.8±9.5 |

Table 8: Performance of the data-driven PGAP adjustment on highly dense graphs (d=60, average degree 24). SHD is reported.

| Noise | Graph&Edge | ER & 12d | | SF & 12d | |
| | Prior Proportion | 0.2 | 0.4 | 0.2 | 0.4 |
|---|---|---|---|---|---|
| Exp | NOTEARS-Softmax | 648.8±8.9 | 601.2±20 | 517.8±18 | 367.3±8.5 |
| | +PGAP(median) | 617.8±7.7 | 541.7±18 | 473.7±20 | 351.5±9.7 |
| | +PGAP(75th) | 620.3±7.5 | 548.0±18 | 472.7±18 | 346.0±10 |
| Gauss | NOTEARS-Softmax | 650.0±8.8 | 601.0±15 | 513.5±16 | 376.8±15 |
| | +PGAP(median) | 618.3±8.3 | 547.8±14 | 485.3±12 | 356.0±10 |
| | +PGAP(75th) | 627.5±7.4 | 549.7±13 | 478.8±14 | 353.0±11 |
| Gumbel | NOTEARS-Softmax | 647.7±7.2 | 594.3±17 | 511.8±14 | 385.8±15 |
| | +PGAP(median) | 613.2±8.3 | 541.3±16 | 480.7±13 | 345.3±15 |
| | +PGAP(75th) | 619.3±6.9 | 548.7±17 | 475.3±13 | 344.0±11 |

clustering coefficients, characteristic of social and biological networks, where triangular patterns may appear more frequently. The Stochastic Block Model (SBM) creates graphs with community structures by varying edge probabilities within and between node blocks, causing our target patterns to be unevenly distributed. Finally, Geometric Random Graphs (GRG) connect nodes based on their proximity in a metric space, resulting in spatially-driven topologies.

The experimental settings for these models were varied to produce graphs with different densities and properties. For WS, we adjusted the lattice dimension; for SBM, the number of blocks; and for GRG, the connection radius. Tables 9, 10, and 11 present the performance of NOTEARS-Softmax (NT-Soft) with and without PGAP enhancement for the WS, SBM, and GRG models, respectively.

The results shown across these tables demonstrate that PGAP consistently improves both F1 and SHD metrics across all three graph models and their varying parameter settings. This robust performance is particularly notable because these models generate topologies where the natural frequency and distribution of triangular and collision patterns can differ significantly from standard ER or SF graphs. This provides strong evidence that the deviation mechanism identified in our work is a fundamental phenomenon and that PGAP's pattern-guided approach is an effective and generalizable strategy for improving prior integration.

# E   Limitation and Further Discussion

This section provides further discussion on key aspects of the PGAP framework and its evaluation. We elaborate on the interpretation of edge weight deviation ($\gamma$), the applicability of our analysis to different scoring functions, the assumptions and generalizability of our theoretical findings, the robustness of the identified structural patterns, considerations regarding prior knowledge quality,

Table 9: Performance of PGAP on WS graphs. (·) indicates the number of edges.

| Nodes | Parameter | NT-Soft | | +PGAP | |
|---|---|---|---|---|---|
| | | F1 | SHD | F1 | SHD |
| 30 | Lattice=1 (60) | 53.5±5.0 | 50.7±6.6 | 56.5±4.7 | 44.8±6.3 |
| | Lattice=2 (147) | 50.6±3.6 | 137.7±9.7 | 54.5±4.3 | 107.0±8.2 |
| 60 | Lattice=1 (120) | 56.8±1.4 | 104.7±4.9 | 58.8±1.5 | 93.3±6.0 |
| | Lattice=2 (333) | 46.9±2.0 | 345.7±11.2 | 50.5±2.5 | 290.5±15.5 |

Table 10: Performance of PGAP on SBM graphs. (·) indicates the number of edges.

| Nodes | Parameter | NT-Soft | | +PGAP | |
|---|---|---|---|---|---|
| | | F1 | SHD | F1 | SHD |
| 30 | Block=5 (71) | 57.1±3.0 | 55.3±4.2 | 58.0±6.5 | 49.8±5.7 |
| | Block=10 (64) | 58.9±5.4 | 44.5±3.0 | 58.3±8.4 | 44.8±9.3 |
| 60 | Block=5 (118) | 56.0±8.4 | 98.3±9.9 | 57.1±5.3 | 83.5±7.6 |
| | Block=10 (118) | 56.9±3.5 | 90.3±9.3 | 62.5±2.5 | 73.8±3.2 |

Table 11: Performance of PGAP on GRG. (·) indicates the number of edges.

| Nodes | Parameter | NT-Soft | | +PGAP | |
|---|---|---|---|---|---|
| | | F1 | SHD | F1 | SHD |
| 30 | Radius=0.2 (43) | 61.1±3.2 | 26.0±2.8 | 62.9±3.9 | 23.0±3.3 |
| | Radius=0.3 (101) | 62.1±1.7 | 69.5±3.4 | 63.2±1.5 | 62.7±3.1 |
| 60 | Radius=0.2 (195) | 55.5±1.3 | 159.5±6.5 | 56.2±2.9 | 148.0±7.9 |
| | Radius=0.3 (368) | 56.7±1.8 | 280.2±13.6 | 57.5±3.5 | 257.8±8.9 |

the positioning of PGAP within the landscape of prior integration methods, and its computational complexity.

**On the Interpretation and Magnitude of Deviation**   Section 4.1 of the main text introduces the critical concept of edge weight deviation. This phenomenon arises when qualitative prior knowledge is integrated into the learning of quantitative SEMs, potentially causing learned edge weights to differ from those purely indicated by the data. Our analysis specifically examined how a localized deviation, defined as $\gamma = \hat{W}_{\beta,j} - W^*_{\beta,j}$ (representing the difference between a prior-influenced weight $\hat{W}_{\beta,j}$ and the OLS-optimal weight $W^*_{\beta,j}$) can propagate and influence other weights. This appendix offers a more detailed exposition of this deviation and its broader implications.

The theoretical framework presented (Lemmas 2 and 3) demonstrates that such a deviation $\gamma$ on a prior-enforced edge can linearly affect the weights of other potential incoming edges to the same target node. When this deviation $\gamma$ is substantial, it can lead to the erroneous formation of spurious edges, where $W_{k,j}$ becomes non-zero even if no true direct relationship $X_k \rightarrow X_j$ exists. As further elaborated in Section 4.2, these spurious edges, in combination with the prior-influenced edge and other true connections, result in the characteristic "triangle" and "double collision" graph patterns. Consequently, these patterns serve as observable structural indicators within the learned graph, signaling that significant weight deviations, driven by the enforcement of prior knowledge, may be actively distorting the local structure being learned around a particular node.

While the deviation $\gamma$ discussed in Section 4.1 is a localized mechanism observed within an optimization step, its repeated occurrence due to strong or misaligned prior enforcement is what contributes to a more significant, global issue: the final learned edge weights in the SEM ($\tilde{W}_{ij}$) can diverge considerably from the true, underlying weights of the data-generating process ($B_{ij}$). This ultimate discrepancy represents a core challenge when attempting to bridge qualitative priors with the quantitative nature of SEMs. The magnitude of this deviation from the true SEM weights is influenced

by several factors. A primary factor is the inherent inconsistency between the qualitative prior and the actual quantitative strength and nature of the relationship in the true data-generating system. Furthermore, the specific strategy used to integrate prior knowledge, be it hard constraints on weight values or soft constraints via penalty terms in the loss function (as discussed in Section 2), heavily dictates how strongly the learning process adheres to the prior versus fitting the observational data. The 'strength' or weighting assigned to these prior-related terms in the objective function acts as a direct modulator of this critical balance. If prior enforcement is too rigid, it can overshadow the evidence present in the data, leading to learned weights that, while satisfying the prior, poorly reflect the true underlying causal mechanisms.

Although the precise threshold for the deviation $\gamma$ to induce the aforementioned spurious patterns is data-dependent (influenced by aspects such as noise levels and the magnitudes of other true edge weights), these patterns become increasingly probable as $\gamma$ grows substantial. This typically happens when the prior-enforcement mechanism compels an edge weight to a value markedly different from what local data evidence among the current set of predecessors would otherwise support. It often reflects a situation where a qualitative statement (e.g., "an edge exists") is translated into a quantitative target that is excessively strong or misaligned with the underlying reality of the SEM. The PGAP framework is designed to leverage these observable structural patterns as signals of such problematic weight deviations. By detecting these patterns, PGAP adaptively adjusts the prior integration process (detailed in Appendix C) to mitigate these deviations, aiming for a more accurate and robust estimation of both the graph structure and its associated edge weights.

**Computational Complexity of PGAP**   The PGAP framework, as detailed in Appendix C (Algorithms 1 and 2), involves a two-stage learning process. First, a base structure learning algorithm is run to produce an initial graph, $W_{stage1}$. Second, after detecting specific structural patterns in $W_{stage1}$ and adaptively adjusting either prior loss weights (for soft constraints) or local regularization strengths (for hard constraints), the base algorithm is run again to yield the final graph, $W_{final}$.

The primary computational load thus consists of two executions of the chosen base continuous optimization learning algorithm. The intermediate pattern detection step, which involves checking local neighborhoods around each of the $|\Pi|$ prior edges for specific triangle or double collision patterns, is efficient. For a graph with $d$ nodes, the complexity of this pattern search is generally much lower than that of a full run of the base learning algorithm.

Consequently, if the base algorithm has a computational complexity of $T_{base}$, the overall complexity of PGAP is approximately $2 \times T_{base}$, plus the minor overhead of pattern detection. PGAP therefore maintains the same asymptotic complexity class as the underlying base structure learning method.

**Applicability of PGAP to the Negative Log-Likelihood Scoring Function**   The theoretical framework underpinning PGAP is developed based on the observation that optimizing common scoring functions like least squares ($Q_{ls}$) often decomposes into node-specific OLS problems. This appendix briefly discusses the direct applicability of this framework when using the Negative Log-Likelihood scoring function, commonly employed for SEMs.

The NLL scoring function for a SEM, as presented in the Preliminaries (Eq. 2 using $Q_{nll}$), aims to find the weight matrix $W$ that maximizes the likelihood of the observed data. During the continuous optimization process, an acyclicity constraint $h(W) = 0$ is enforced, which implies a topological ordering of the variables for any candidate $W$. Given this ordering, the estimation of the weights of incoming edges for any particular node $X_j$ from its set of candidate parent nodes $t_j$ effectively reduces to a linear regression problem.

$$\min_W Q_{nll}(W; \mathbf{X}) \Leftrightarrow \min_W \sum_{i=1}^{d} \log \|\mathbf{X} - \mathbf{X}W\|_2^2 \Leftrightarrow \min_{W_i} \|\mathbf{X} - \mathbf{X}W_i\|_2^2 \quad i \in [1, 2, ..., d] \quad (62)$$

This is the same per-node optimization problem as considered for the $Q_{ls}$ score in Section 4.1.

**Assumptions and Generalizability of Theoretical Findings**   Our theoretical framework, while utilizing assumptions like linear SEMs and faithfulness for analytical tractability (as discussed in Section 4.1 and Appendix B), builds upon the identification of local graph patterns (triangle and double collision). As noted in Section 4.2, the conditions in Lemma 3 for pattern formation depend on local structures (direct connections or common children paths). This suggests that the core mechanism

of deviation-induced pattern formation might still provide valuable insights even when some global model assumptions are relaxed. Future work could explore extending the theoretical analysis to non-linear models or other graph families, building on preliminary investigations such as those in Appendix D regarding non-linear scenarios.

**Robustness and Specificity of Deviation-Induced Patterns**    While our theoretical results (Lemmas 4 and 5) suggest that the identified triangle and double collision patterns are expected to be significantly more prevalent when substantial prior-induced deviation occurs compared to their natural occurrence in typical sparse graphs, they may not be exclusively caused by this phenomenon. PGAP, therefore, uses their detection in conjunction with a known prior-enforced edge primarily as a heuristic signal that such deviation is likely problematic and warrants adaptive adjustment. Investigating a broader range of structural signals or more sophisticated detection mechanisms could be a direction for future enhancements.

**Handling of Prior Knowledge Quality**    If a significant portion of the prior knowledge incorporated into the learning process is fundamentally incorrect, PGAP's mechanism (reducing the influence of priors involved in the targeted deviation patterns) might inadvertently down-weight these incorrect priors if they lead to such structural distortions. In this sense, it could offer some resilience against grossly misspecified priors by reducing their impact. However, PGAP is not explicitly designed as a prior validation or correction tool. Its main goal is to rationalize the use of priors assumed to be qualitatively valuable but potentially quantitatively mis-specified, a common assumption as noted in Section 3.

**Positioning Relative to Proactive Prior Integration Strategies**    PGAP is designed as an adaptive framework that can augment a wide range of existing structure learning algorithms and prior integration techniques (both 'hard' and 'soft' constraint methods, as demonstrated in our experiments, e.g., in Section D.2 and related tables in Appendix D). Its strength lies in identifying and mitigating a specific, theoretically grounded adverse effect that can occur even with established prior enforcement methods. While developing more sophisticated upfront prior integration methods (e.g., learning prior strengths or incorporating prior uncertainty directly) is a valuable and active area of research, PGAP offers a complementary approach. It addresses an observable consequence of potential mismatches between qualitative priors and quantitative models, effectively acting as a refinement layer. The two-stage process allows for an initial assessment of prior impact, followed by a targeted adjustment. Future work could explore synergies between PGAP's pattern-guided adaptation and more complex prior modeling techniques. For example, the structural signals detected by PGAP could potentially inform the learning of prior confidence scores or guide more nuanced regularization schemes in a unified framework.

**On the Linear Assumption and Generalization to Nonlinear Models**    Our formal theoretical analysis is developed within the linear SEM framework. This focus is deliberate. Linear SEMs are a cornerstone of causal modeling, not only used extensively across scientific and industrial domains but also serving as a critical object of study for developing more advanced methods. The continued prevalence of linear models is largely due to their interpretability, where linear coefficients provide a clear measure of causal influence, and their computational tractability. In many complex systems, particularly in high-dimensional, low-sample-size settings, linear models serve as powerful first-order approximations where fitting complex nonlinear models is often statistically infeasible and prone to overfitting. Addressing the unresolved problem of robustly integrating imprecise prior knowledge within this foundational domain is therefore a significant contribution.

Although extending our formal analysis to general nonlinear models is a challenging task, the core principle of the PGAP framework is generalizable. Our path for such an extension begins by generalizing our core concepts from the parametric to the nonparametric domain, leveraging the framework of Zheng et al. [2020]. The concept of an edge weight is replaced by a functional dependency measure, and "edge weight deviation" becomes a more general "functional dependency deviation."

To understand how this functional deviation propagates, we then adopt a local perspective. When a prior introduces a functional deviation, the learned function $f_j$ and the data-optimal function $f_j^*$ are no longer identical. In a linear model, this difference is inherently global. In the nonlinear case, however, the two functions $f_j$ and $f_j^*$ can be identical over large areas and only diverge in specific

local regions of the input space. It is within these regions of divergence that we can analyze the propagation mechanism. Here, the behavior of $f_j$ can be approximated by its first-order Taylor expansion, rendering it locally linear. The partial derivatives of $f_j$ act as the coefficients of this local linear approximation. When pressure from the prior forces one of these "local coefficients" to be non-optimal, the optimization process is incentivized to adjust other local coefficients to minimize the loss. This local mechanism is analogous to the error propagation we formally analyzed in the linear setting. The adjustment of other local coefficients can cause the global functional dependency measure to become non-zero, signifying the creation of a spurious edge and leading to the formation of the same topological patterns we leverage in PGAP.

This local perspective also provides a crucial insight into why the effects of deviation might appear less pronounced in some nonlinear settings. If a local region where the prior causes significant deviation is sparsely sampled, its contribution to the global loss will be minimal, and the optimizer may find a solution that largely ignores the prior's influence in that region. This does not mean the phenomenon is negligible. On the contrary, it points to an often overlooked aspect of causal discovery. While global dependency metrics are standard, they can average out important local effects that occur only in specific regions of the input space. We believe that analyzing and accounting for such local prior-induced deviation represents a key step forward for the field, especially for emerging applications such as discovering context-specific causal effects or understanding heterogeneous treatment effects in subgroups.

