# OpenReview forum: "Pattern-Guided Adaptive Prior for Structure Learning"
_NeurIPS.cc/2025/Conference — NeurIPS 2025 poster_

### Official Review · Reviewer_xCbJ · 2025-06-22

**Clarity:** 3
**Significance:** 3
**Originality:** 3
**Rating:** 5
**Confidence:** 3

**Summary:**

The paper studies the effect of using imprecise prior in structure learning for linear DAG when the prior knowledge supplied by domain experts is misspeficied. The paper shows enforcing such priors can create edge weight deviation, which then propagates and spawns spurious edges. Two structural patterns are identified: triangles and double collisions. The expected number of apearance of such patterns is derived. This observation is used in a two-stage framework called Pattern-Guided Adaptive Prior (PGAP). PGAP detects the patterns after an initial run of any differentiable methods, then adaptively down-weights the priors before a second optimization pass. Extensive synthetic benchmarks and experiments are conducted to support the proposal.

**Questions:**

- The proposed framework is based on heuristics. Is it going to hurt the performance if the prior is correctly specified? More formally, are the identified stuctural patterns sufficient and necessary conditions for imprecise priors?
- In the experiments, what are the priors and how are they specified? Are they precise or imprecise?

**Ethical Concerns:**

["NO or VERY MINOR ethics concerns only"]

**Final Justification:**

The resposne from the authors addressed my questions on priors and experiments. I keep my score unchanged and am inclined to acceptance.

**Limitations:**

Limitations are discussed in Appendix E.

**Quality:**

3

**Strengths And Weaknesses:**

**Strength:**

- The paper provides a fresh and quantitative viewpoint on using priors during structure learning. The observations come from solid theoretical analysis of the consequence of imprecise priors.
- PGAP is a light wrapper around any continuous DAG learner thus has algorithmic simplicity. The pattern detection is local and cheap.
- The experiments are comprehensive and support the effectiveness of the proposed framework and show consistent empirical benefits.

**Weakness:**

- One of the main contributions is the proposal of PGAP framework. However, it is only briefly described in words in the main paper. The full details are burried in the appendix. It is unclear how does the framework work by solely looking at Section 4.3.
- The proposed framework is inspired by the observation from analysis, while is based on heuristics. There is not consistency guarantee of the framework, see questions.
- The theoretical analysis of deviation relies on linear SEM thus might not extend to general DAG model. Although some exploratory experiments for nonlinear models are given in the appendix.

---

> ### Author Rebuttal · Authors · 2025-07-28
>
> We are very grateful to you for your positive assessment and supportive comments on our work. We sincerely appreciate that you found our viewpoint to be "fresh and quantitative," our analysis "solid," and our experiments "comprehensive." We will address the points you raised to further improve the paper and would be happy to answer any further questions you might have.
>
> Before we address the specific questions, we would like to clarify a key distinction between "accurate'' and "precise'' priors, as this concept is fundamental to both of the reviewer's questions and to the core motivation of our paper. In our work, and in the broader literature, prior knowledge is typically assumed to be accurate. This means the prior correctly asserts the existence of a true relationship, for example that the edge $X_i \rightarrow X_j$ exists in the ground truth graph. Our experiments strictly adhere to this principle by using only accurate priors. However, such priors are rarely precise because they do not specify the exact quantitative strength of that relationship. This strength is the numerical edge weight that a quantitative model like an SEM must learn. The central challenge we address stems from this gap between a qualitatively accurate assertion and the quantitatively precise value required by the model.
>
> We now answer the specific questions raised.
>
> **Q1:**
> We thank you for this insightful question regarding PGAP's behavior with a correctly specified prior. Our framework is indeed designed not to be negatively impacted in this ideal scenario. A truly "correct" prior (one that is both accurate and precise) would result in zero deviation ($\gamma=0$). In this ideal case, our theory shows that no error propagation will be induced, and PGAP's mechanism for detecting special patterns is rarely triggered. We will clarify this important point in the revised manuscript to ensure this aspect of our method is clear.
>
> Regarding the relationship between the patterns and imprecise priors, while it is not a strictly necessary and sufficient condition, our paper's core contribution is to establish and leverage the strong, fundamental connection between them. Our argument is built on a three-part foundation of theoretical proof, expectation analysis, and empirical validation. Specifically, our analysis in Lemmas 2 and 3 provides a formal proof that imprecise priors, when they cause edge weight deviation, directly lead to the formation of these two specific patterns. Building on this, our expectation analysis in Lemmas 4 and 5 addresses their utility as a reliable signal. While it is possible for these patterns to occur naturally, our analysis demonstrates that their appearance is a rare event compared to their prevalence when induced by deviation. This makes their detection a strong indicator of a problematic prior integration. Finally, this theoretical linkage is robustly validated through empirical validation. Our experiments consistently show that adaptively responding to these structural signals leads to significant performance improvements, confirming their practical utility. Therefore, using these patterns as a signal is a theoretically-grounded and empirically-validated strategy for identifying errors caused by imprecise priors.
>
> **Q2:**
> In all our experiments, the priors were generated by randomly selecting a proportion of edges from the true, data-generating DAG. Therefore, all priors used are accurate because they correspond to a true edge in the ground truth graph.
>
> However, a key point of our work is to distinguish this from being precise. While an accurate prior correctly identifies the existence of a causal link, it does not specify the exact numerical weight of this relationship, which is the quantitative value the SEM aims to learn. In our experiments, the prior is incorporated as a penalty that encourages a non-zero weight, thus satisfying the qualitative, accurate prior. But because the exact target weight is unknown and unspecified, this enforcement is inherently imprecise. This setup directly models the common real-world scenario where domain knowledge can confirm a causal link but not its precise strength, which is the central challenge our paper addresses.
>
> ***
>
> **Regarding the Weaknesses Mentioned:**
>
> * On the clarity of the PGAP framework's description in the main paper, we agree that Section 4.3 is too brief. We will revise the main text to include a more detailed description of the two-stage procedure and the core mechanics of the adaptive adjustment.
> * As discussed in our response to Q1, our work focuses on establishing the strong, fundamental connection between imprecise priors and specific structural patterns, which istheoretically-grounded, and empirically-validated.
> * Finally, regarding the theory's reliance on linear SEMs, we have addressed this in detail in our response to **Reviewer 7Uc5, Q2**, where we outline a concrete path for generalization.
>
> We hope these clarifications are helpful, and we thank the reviewer again for their encouraging and constructive feedback.
>
> As a final point, we wish to share our response to another reviewer regarding the diversity of graph models. We conducted new experiments at their suggestion, but due to strict space constraints, we are unable to include the full analysis in that rebuttal. As these results strongly support our paper's claims of robustness and generalizability, we feel it is important to share this analysis with you as well. We have appended that full response below for your consideration and hope you find it informative.
>
> These experiments generate random DAGs using three distinct models. The Watts-Strogatz (WS) model simulates social and biological networks with high clustering coefficients and short path lengths, where triangular patterns appear more frequently than in a general ER graph. The Stochastic Block Model (SBM) creates a non-uniform, blocky connection pattern by dividing nodes into different blocks, causing our target patterns to be unevenly distributed. Finally, Geometric Random Graphs (GRG) are formed by connecting nodes based on their proximity in a metric space.
>
> The experimental settings are as follows: for WS, the lattice dimension $k \in \{1,2\}$ and rewiring probability $p=0.3$; for SBM, the number of blocks $k \in \{5,10\}$ and inter/intra block edge probability $p=0.1$; for GRG, the connection radius $k \in \{0.2,0.3\}$. The experimental results under these diverse underlying graphs are shown in Table 1.
>
> **Table 1: Performance of PGAP in diverse underlying graphs and critical parameter settings. In the 'Lattice', 'Radius', and ’Block‘ column, (·) represents the number of edges under the node d and the critical parameter k.**
>
> | Graphs |             |          |     WS     |              |                |              |              |     SBM      |              |              |               |          |    GRG     |              |               |
> | :----: | :---------: | :------: | :--------: | :----------: | :------------: | :----------: | :----------: | :----------: | :----------: | :----------: | :-----------: | :------: | :--------: | :----------: | :-----------: |
> | Method |             | NT-Soft  |            |    +PGAP     |                |              |   NT-Soft    |              |    +PGAP     |              |               | NT-Soft  |            |    +PGAP     |               |
> |  Node  | **Lattice** |    F1    |    SHD     |      F1      |      SHD       |  **Block**   |      F1      |     SHD      |      F1      |     SHD      |  **Radius**   |    F1    |    SHD     |      F1      |      SHD      |
> |   30   | 1 **(60)**  | 53.5±5.0 |  50.7±6.6  | **56.5±4.7** |  **44.8±6.3**  |  5 **(71)**  |   57.1±3.0   |   55.3±4.2   | **58.0±6.5** | **49.8±5.7** | 0.2 **(43)**  | 61.1±3.2 |  26.0±2.8  | **62.9±3.9** | **23.0±3.3**  |
> |        | 2 **(147)** | 50.6±3.6 | 137.7±9.7  | **54.5±4.3** | **107.0±8.2**  | 10 **(64)**  | **58.9±5.4** | **44.5±3.0** |   58.3±8.4   |   44.8±9.3   | 0.3 **(101)** | 62.1±1.7 |  69.5±3.4  | **63.2±1.5** | **62.7±3.1**  |
> |   60   | 1 **(120)** | 56.8±1.4 | 104.7±4.9  | **58.8±1.5** |  **93.3±6.0**  | 5 **(118)**  |   56.0±8.4   |   98.3±9.9   | **57.1±5.3** | **83.5±7.6** | 0.2 **(195)** | 55.5±1.3 | 159.5±6.5  | **56.2±2.9** | **148.0±7.9** |
> |        | 2 **(333)** | 46.9±2.0 | 345.7±11.2 | **50.5±2.5** | **290.5±15.5** | 10 **(118)** |   56.9±3.5   |   90.3±9.3   | **62.5±2.5** | **73.8±3.2** | 0.3 **(368)** | 56.7±1.8 | 280.2±13.6 | **57.5±3.5** | **257.8±8.9** |

---

> > ### Comment · Reviewer_xCbJ · 2025-08-05
> >
> > I thanks the authors for their detailed reply. I keep my positive score uchanged.

---

> > > ### Author Response · Authors · 2025-08-06
> > >
> > > We are very grateful for your positive assessment and supportive comments. Your recognition of our work is a great encouragement to us.
> > >
> > > The constructive suggestions are also greatly appreciated. Based on the feedback, we will move the core algorithmic details from the appendix to the main text to enhance clarity. We will also further clarify the distinction between "accurate" and "precise" priors to address the questions raised.
> > >
> > > Thank you again for the valuable feedback and the confidence shown in our work.

---

### Official Review · Reviewer_ETuv · 2025-06-30

**Clarity:** 3
**Significance:** 3
**Originality:** 4
**Rating:** 5
**Confidence:** 4

**Summary:**

Under the assumption of a linear model and causal sufficiency, this paper studies how integrating prior knowledge (edge existence) into continuous optimization-based causal discovery methods may lead to errors in the returned graph. They show that if an initial edge weight is estimated incorrectly, this error can propagate to other edge weights. They conclude that this error will show up in the form of spurious edges, and define two graphical structures (triangular collision and double collisions) that may occur artificially. Finally, they develop a sub-routine (PGAP) which can be added on top of any continuous optimization-based method, which adaptively lowers the weight placed on prior knowledge when the number of triangular/double collision patterns exceeds a threshold. They experimentally validate the efficacy their sub-routine in combination with a few standard continuous opt. methods in both synthetic and real data, showing that it almost uniformly increases performance, although by different margins in different settings.

**Questions:**

Q1 Can you provide a principled justification for focusing exclusively on triangular and double collision patterns, and characterize what proportion of weight deviation errors these patterns capture?

Q2 How does PGAP perform in dense graphs without oracle hyperparameter selection, and can you provide a practical tuning procedure for α?

Q3 Can you outline a concrete path for extending PGAP beyond linear models, or provide theoretical analysis showing why such extension is fundamentally difficult?

I'm happy to increase my rating if the authors adequately address my concerns, both in this Questions section and as well in the Strengths and Weaknesses section.

**Ethical Concerns:**

["NO or VERY MINOR ethics concerns only"]

**Final Justification:**

The authors addressed my concerns, particularly by providing a data-driven way to set the threshold used to determine whether the graph has "too many" collisions. I recommend to accept.

**Limitations:**

See 'Clarity' and "Quality" section of the Strengths and Weaknesses section.

**Paper Formatting Concerns:**

None.

**Quality:**

3

**Strengths And Weaknesses:**

Strengths:

​Originality - Discovery methods tend to be sensitive to error propagation, and often return results that seem unlikely, prima facie. The subroutine PGAP suggested in this paper can be seen as a sort of "self-correcting" mechanism, which adjusts the optimisation function (in this case lowering the weight on the prior constraints) used to derive the causal structure whenever the graph structure suggested by the algorithm seems unlikely (# of triangle/double collisions is high). To my knowledge, explicit analysis and development of error-correcting sub-algorithms remains relatively unexplored in the discovery literature. Additionally, the way the authors reason from the occurence of a local error  (a "weight deviation") and consider the resulting cascade of global errors it might impart helps deepen our explicit understanding of error propagation in discovery methods.

Weaknesses:

Clarity - Although the general organization of sections in the paper is fair, key information is often times not made explicit, or is deferred. For example, although this paper develops PGAP explicitly for causally sufficient linear models, leveraging the linearity assumption throughout the analysis, this functional assumption is not made clear to the reader until the middle of page 3 (Section 3). Ideally, an assumption this foundational to the methodology should be present in the title, abstract, and/or contribution part of the introduction. In Section 4.3, the authors defer the pseudocode description of their algorithm PGAP entirely to the Appendix, describing their proposed subroutine across only two sentences. This hinders the reader in understanding the author's proposed idea, and obscures the contribution of their work. Another example is in Section 6, where the authors defer all discussion of the limitations of their PGAP framework to the appendix, while the summary they supply does not provide any detail as to the shortcoming of their approach.

Quality - Although the claims presented in the main text appear to be correct, and supported by the empirical evidence provided, it is not clear that the core idea by the authors is sufficiently developed to warrant a recommendation for publication. In Section 4.1, the authors characterize, in general, how weight deviations can propagate and affect learned weights of other edges (Lemma 2, Eq 11, etc,); however, in Section 4.2, they only explore one set of possible graphical structures that result from errors described in 4.1, namely the triangular and double collision patterns. Theres no explanation given why we should focus on these collision patterns, as opposed to any other set of structures that could arise. Although the initial analysis is interesting, it seems somewhat arbitrary that the authors chose to focus on only a pair of possible structures, without any compelling motivation.

Significance - Although the core idea of adaptive error-correcting subroutine (PGAP) is compelling, the much of this paper's analysis, both theoretical and empirical, seems to be of limited significance to the broader field. For example, they tackle error-correction only from the perspective of continuous-optimization based causal discovery methods, which have been uniquely criticized (among discovery methods) for being especially sensitive to unequal noise variances, performing poorly after standardization, and failing to generalize well to nonlinear and non-convex settings [1]. Additionally, the entirety of the analysis here remains limited to the linear setting, without discussion on how the error-correcting strategy could be generalized (although, it does not seem obvious to me that the approach here would generalize). Finally, even though the authors do show that PGAP does improve performance in many settings, it seems that both the theoretical and empirical results point to the utility of PGAP largely in sparse settings. Theoretically, in sparse settings relatively few collisions will occur, so adaptively penalizing the prior when collisions occur should be expected to improve performance. However, in dense settings, one would expect many more collisions, and thus penalizing might fail to improve performance. Figures 16 and 17 in the Appendix reflect this - in sparse graphs (Figure 16), NoTears+PGAP outperforms NoTears for almost all choices of the adaptive reduction parameter $\alpha$. However, for dense graphs (Figure 17) NoTears+PGAP only outperforms NoTears for specific values of $\alpha$, with the range being narrower when there is a lower amount of prior knowledge. Therefore, tuning of the hyperparameter $\alpha$ seems somewhat critical to the performance of PGAP in dense settings with limited priors. Unfortunately, the authors do not suggest a tuning method, opting instead to use the optimal parameter here throughout the rest of the experiments, limiting the general applicability of PGAP to real world settings.

---

> ### Author Rebuttal · Authors · 2025-07-27
>
> We sincerely thank you for your thoughtful and constructive feedback. We have provided detailed answers to your comments and would be happy to answer any further questions you might have.
>
> **Weaknesses (Clarity):** To improve the clarity of the paper, we will revise the manuscript to be more self-contained by moving the discussion of the foundational linear model assumption, the PGAP algorithm details, and the core limitations from the appendix into the main text.
>
> **Weaknesses (Quality) & Q1:**
> Our focus on the triangular and double collision patterns is not an arbitrary choice, but stems directly from our theoretical analysis of the fundamental error propagation mechanism, as presented in Lemma 2 and 3. Lemma 2 establishes that a deviation on a prior-enforced edge can propagate to induce an erroneous weight on another potential edge, while Lemma 3 precisely characterizes the necessary and sufficient conditions for this propagation to occur. These conditions, which relate the two potential parent nodes $X_k$ and $X_\beta$ that both point to $X_j$, are precisely: 1. A direct edge exists between $X_k$ and $X_\beta$ in the true graph, or 2. $X_k$ and $X_\beta$ share a common child node in the true graph.
>
> These two conditions constitute the exhaustive set of elementary mechanisms by which a deviation on a single prior-enforced edge directly induces an error on one other edge. We term this fundamental, one-step process first-order error propagation. The triangular and double collision patterns are, in turn, the direct structural manifestations that arise when these pre-existing structure are combined with the spurious edge $(X_k \rightarrow X_j)$ induced by the deviation. Therefore, we focus on them because they are the elementary building blocks of error arising from our theoretical model, not arbitrary heuristics. Furthermore, since any more complex error structure would be composed of these elementary patterns, analyzing them is the most logical and principled first step.
>
> Regarding the proportion of errors these fundamental patterns capture, they are exhaustive for the most elementary error type, capturing 100% of these cases. This is because within our theoretical analysis of a single deviation, Lemma 3 shows that these two patterns are the only possible structural outcomes.
> As for more complex error structures that may arise from multiple interacting deviations, providing a single universal percentage of all errors captured is theoretically challenging. However, our work demonstrates this is not necessary because these two fundamental patterns already serve as a dominant and reliable signal of error. Firstly, our expectation analysis (Lemmas 4 and 5) shows their frequency when induced by deviation far surpasses their natural rate of occurrence, making them a strong theoretical indicator. Secondly, the consistent performance gains across our experiments validate their practical power. The fact that targeting these patterns is a highly effective strategy strongly suggests they are responsible for a sufficiently large and impactful proportion of the total errors.
>
> **Weaknesses (Significance) & Q2:**
> In fact, our experiments also compare with the state-of-the-art method NT-LogLL [1], which addresses sensitivity to unequal noise variances. The results show PGAP consistently enhances even this modern and robust algorithm.
>
> Regarding the performance in dense graphs, our initial implementation reduces the prior's influence based on the raw count of detected patterns. As our experiments show, this is a highly effective strategy in the sparse settings common to many causal discovery problems. The core challenge in denser graphs is the higher baseline number of naturally occurring triangular and double collision patterns. A simple count-based adjustment therefore risks over-penalizing priors by reacting to structurally necessary patterns as if they were errors.
>
> Fortunately, our theoretical framework allows for a straightforward and elegant refinement. For dense graphs where a high number of patterns occur naturally, the signal can be extended from the raw pattern count to the number of excess patterns that emerge above the graph's natural structural baseline. In the sparse settings, this natural baseline is negligible or near-zero. Consequently, the observed raw pattern count serves as a direct and effective proxy for the excess count, which explains why our initial implementation is so effective in those scenarios. For denser graphs where this simplification is no longer sufficient, our framework allows us to refine the procedure to explicitly isolate the true signal. This leads to the following practical and fully data-driven tuning procedure:
>
> The procedure begins by learning an initial graph $W_{stage1}$. The key refinement is how we interpret the pattern counts. Instead of reacting to raw counts, we first establish a data-driven baseline. We iterate through every edge in the learned graph $W_{stage1}$ and count the patterns associated with each edge. This gives us an empirical distribution of pattern counts specific to the lea. From this distribution, we define the "normal" level of patterns ($n_{baseline}$) using a robust statistic like the median or the 75th percentile. For a prior-enforced edge $\pi$, we then compute its excess pattern count: $n_{excess, \pi} = \max(0, n_{observed, \pi} - n_{baseline})$. The strength of the adjustment (SA) is made adaptive by normalizing this excess count against the maximum count observed in the graph ($n_{max}$):
> $$\text{SA} _{\pi}=\frac{n _{excess, \pi}}{ n _{max} - n _{baseline} +1}$$
>
> This factor, representing the anomaly score of the pattern count, can then be used in a straightforward update rule, such as $\alpha_{\pi, new} \leftarrow \alpha_{\pi, old} \times (1 - \text{SA}_{\pi})$.
>
> This refined procedure is a natural extension of our theory. It is robust to graph density because the baseline is empirically derived from the graph's own structure. Furthermore, it is a non-oracle method, as both the signal and the adjustment strength are determined automatically from the data. This demonstrates the flexibility of the PGAP framework and its straightforward applicability to the practical challenges.
>
> To empirically validate this proposed non-oracle procedure, we have already conducted a set of preliminary experiments, the results of which are presented in the accompanying table. The results consistently demonstrate that both PGAP variants, using either the median or the 75th percentile to establish a data-driven baseline, achieve a significant performance improvement over the standard method, with this robust improvement holding across challenging dense graph scenarios (d=40 with 8d edges, i.e. average degree = 16). This empirical evidence affirms that our proposed refinement is not only theoretically sound but also a practical and effective solution.
>
> **Table 1:** Performance of PGAP in the improved adjustment scenario based on empirically normalized adaptive.
>
> |        |          Node          |              |     d=20     |              |              |               |     d=40      |               |               |
> | :----: | :--------------------: | :----------: | :----------: | :----------: | :----------: | :-----------: | :-----------: | :-----------: | :-----------: |
> |        |       Graph&Edge       |    ER&4d     |              |    SF&4d     |              |     ER&8d     |               |     SF&8d     |               |
> | Noise  |    Prior Proportion    |     0.2      |     0.4      |     0.2      |     0.4      |      0.2      |      0.4      |      0.2      |      0.4      |
> |  Exp   |    NOTEARS-Softmax     |   69.2±8.4   |   64.0±6.3   |   54.5±7.7   |   49.2±6.9   |   296.0±7.8   |   259.0±11    |   218.3±0.8   |   180.0±13    |
> |        |     +PGAP(median)      | **62.3±7.9** | **56.7±6.0** | **51.5±8.5** | **42.3±7.3** | **282.0±4.7** | **233.8±7.9** | **213.5±4.6** | **162.8±10**  |
> |        | +PGAP(75th percentile) | **62.2±7.6** | **59.0±6.0** | **52.2±8.2** | **43.2±7.0** | **282.7±5.2** | **231.2±8.3** | **209.2±5.4** | **159.7±11**  |
> | Gauss  |    NOTEARS-Softmax     |   67.8±6.2   |   60.8±4.5   |   53.5±9.2   |   47.3±5.9   |   298.2±7.1   |   258.5±8.0   |   229.5±8.2   |   179.8±8.3   |
> |        |     +PGAP(median)      | **67.5±4.8** | **57.8±6.7** | **47.7±7.8** | **43.3±8.1** | **284.8±4.5** | **238.7±6.8** | **218.5±2.9** | **165.2±9.0** |
> |        | +PGAP(75th percentile) | **66.7±4.5** | **55.7±8.3** | **48.8±9.5** | **43.2±7.0** | **286.8±4.3** | **238.5±8.6** | **214.0±6.1** | **159.2±11**  |
> | Gumbel |    NOTEARS-Softmax     |   71.8±6.1   |   63.3±5.3   |   53.2±8.9   |   47.8±9.4   |   294.5±6.3   |   256.8±7.6   |   222.0±5.9   |   181.5±10    |
> |        |     +PGAP(median)      | **66.3±6.7** | **59.0±6.7** | **47.8±8.5** | **40.0±8.5** | **280.5±6.5** | **238.7±5.9** | **219.0±1.8** | **164.2±9.2** |
> |        | +PGAP(75th percentile) | **64.5±5.5** | **57.7±4.7** | **48.7±7.5** | **40.7±8.3** | **286.8±2.1** | **234.5±8.0** | **217.8±2.8** | **161.8±9.5** |
>
> **Q3:**
> Thank you for this question, which is crucial for understanding the broader applicability of our work. This point was also raised by another reviewer, and we have formulated a detailed response. For the sake of brevity and to avoid repetition, we respectfully refer the reviewer to our response to **Reviewer 7Uc5, Q2**. In that response, we outline a concrete path for extending PGAP to nonlinear models. We discuss how the core concepts of "deviation" and "propagation" can be generalized to a functional domain and argue that the same topological patterns serve as principled signals of error. We also use a local approximation perspective to provide a concrete intuition for the propagation mechanism and discuss why this phenomenon is fundamental to causal discovery.
>
> [1] C Deng, K Bello et.al. Markov Equivalence and Consistency in Differentiable Structure Learning.

---

> ### Author Response · Authors · 2025-08-05
> **Further Clarification on Significance**
>
> Thank you again for your thorough and constructive review. We wish to provide a brief, further clarification on what we believe might be a potential remaining concern regarding the significance of our work, hoping this can fully address your concerns.
>
> Regarding the "Significance" weakness, you noted that continuous-optimization methods have been criticized for several shortcomings [1]. Although the specific reference was not provided, we believe your critique aligns with the important work by Reisach et al. (2021), “Beware of the Simulated DAG!”. That work correctly points out that many such methods perform poorly after data standardization (due to losing "varsortability" signals), with unequal noise variances.
>
> We want to respectfully highlight that a significant part of our experimental design was intended to investigate PGAP's performance under these well-known challenges. This was to demonstrate that PGAP's effectiveness extends beyond the ideal conditions where standard methods work well, and into the very scenarios you highlighted.
>
> Specifically, our submitted manuscript includes the following design considerations and experiments:
>
> 1.  **Performance on Standardized Data**: To avoid the "varsortability" issue, **all our data was standardized** prior to being used in the experiments, as stated on Line 291. This ensures that the performance gains from PGAP are not reliant on the variance patterns identified by Reisach et al., but rather on the true structural properties our theory describes.
>
> 2.  **Robustness to Unequal Noise Variances**: To address the critique of sensitivity to unequal variances, our experimental design explicitly includes this challenging scenario. As shown in **Table 1 (Gauss-NV)** and further in Appendix D.1, PGAP consistently improves performance in various non-equal variance (-NV) settings. Crucially, our results also show that PGAP enhances the performance of **NT-LogLL**, a state-of-the-art baseline specifically designed to be robust against this issue, which demonstrates PGAP's value.
>
> 3.  **Generalization to Nonlinear Settings**: While our core theory is developed in the linear domain for analytical clarity, we conducted exploratory experiments to test PGAP’s applicability beyond this assumption. The results in **Appendix D.6 (Table 6)** show that PGAP continues to offer performance improvements even in nonlinear MLP and MIM settings, validating the generalization potential of its underlying mechanism.
>
> In summary, a key part of our experimental framework was consciously designed to validate PGAP's capabilities in the very scenarios where standard continuous optimization methods are known to falter.
>
> Finally, to further solidify the practical applicability of PGAP in the challenging dense-graph scenarios you pointed out, we have conducted **additional experiments on even denser graphs**. The settings include 60 nodes with an average degree of 24 (corresponding to 12d edges, or a 40% edge density). The preliminary results are shown below:
>
> **Table 2:** Performance of PGAP in the improved adjustment scenario based on empirically normalized adaptive.
>
> |        |      Graph & Edge      |   ER & 12d    |              |   SF & 12d   |               |
> | :----: | :--------------------: | :-----------: | :----------: | :----------: | :-----------: |
> | Noise  |    Prior Proportion    |      0.2      |     0.4      |     0.2      |      0.4      |
> |  Exp   |    NOTEARS-Softmax     |   648.8±8.9   |   601.2±20   |   517.8±18   |   367.3±8.5   |
> |        |     +PGAP(median)      | **617.8±7.7** | **541.7±18** | **473.7±20** | **351.5±9.7** |
> |        | +PGAP(75th percentile) | **620.3±7.5** | **548.0±18** | **472.7±18** | **346.0±10**  |
> | Gauss  |    NOTEARS-Softmax     |   650.0±8.8   |   601.0±15   |   513.5±16   |   376.8±15    |
> |        |     +PGAP(median)      | **618.3±8.3** | **547.8±14** | **485.3±12** | **356.0±10**  |
> |        | +PGAP(75th percentile) | **627.5±7.4** | **549.7±13** | **478.8±14** | **353.0±11**  |
> | Gumbel |    NOTEARS-Softmax     |   647.7±7.2   |   594.3±17   |   511.8±14   |   385.8±15    |
> |        |     +PGAP(median)      | **613.2±8.3** | **541.3±16** | **480.7±13** | **345.3±15**  |
> |        | +PGAP(75th percentile) | **619.3±6.9** | **548.7±17** | **475.3±13** | **344.0±11**  |
>
> These new results, combined with the data-driven tuning procedure we proposed in our main rebuttal, confirm that PGAP maintains its robust performance advantage even under these highly challenging conditions.
>
> We believe these points, combined with our new experimental results, demonstrate that our work is not only aware of the key limitations of prior methods but also shows that our proposed method remains effective when facing them.

---

> ### Comment · Reviewer_ETuv · 2025-08-05
>
> I appreciate the author's thorough response to my review, which has addressed most of my concerns. I will increase my score accordingly.
>
> I recommend that the author's points about how triangular and double collision patterns are the comprehensive set of first order error propagation effects, as well as the new adaptive PGAP (median/75th percentile) method be incorporated into the main text of the paper. These are crucial for understanding the significance of the work.

---

> ### Author Response · Authors · 2025-08-06
>
> We sincerely thank you for reconsidering the evaluation of our work. Your insightful critiques have been invaluable in helping us strengthen the paper.
>
> Based on the valuable suggestions provided, we will incorporate the clarifications on the theoretical justification for our choice of patterns, alongside the new data-driven adaptive procedure, into the main text. We agree this is a crucial step to highlight the significance of the work. To ensure the manuscript is developed to a publishable standard, we stand ready to provide any further clarifications or conduct additional experiments that would be deemed beneficial.
>
> Thank you once again for the invaluable guidance during this review process.

---

### Official Review · Reviewer_7Uc5 · 2025-06-30

**Clarity:** 4
**Significance:** 2
**Originality:** 3
**Rating:** 4
**Confidence:** 3

**Summary:**

This paper centers around Directed Acyclic Graph (DAG) structure learning and aims at addressing the pitfalls of integrating imprecise prior knowledge in the structure learning process, with a particular focus on Structural Equation Models (SEMs). The authors identify the edge weights deviation issue when specifying the edge prior and theoretically analyze how such deviations can propagate through the model, which adversely affects the estimation of other parameters. To mitigate this problem, the authors propose the Pattern-Guided Adaptive Prior (PGAP) framework. PGAP leverages structural signals in the form of two specific graph patterns as reliable indicators, specifically triangular and double collision. As such, PGAP utilizes such patterns during optimization and adaptively adjusts the prior to mitigate the weight deviation. As proof of concept, the authors conduct experiments on synthetic datasets confirming the effectiveness and robustness of PGAP.

**Questions:**

- It would be helpful if the authors could elaborate on the practical scope and applicability of linear SEMs. Specifically, in what types of real- world scenarios are linear SEMs commonly used, and how do the underlying assumptions hold in practice?
- Can the theoretical foundation of the proposed method be extended to accommodate more general, nonlinear models? Given that different structural patterns or deviation mechanisms may emerge in nonlinear settings, it would be valuable to understand the potential for generalization and whether PGAP could be adapted accordingly.
- As described in Algorithms 1 and 2, the prior is obtained through a single round of base structure learning. Could the authors clarify whether there is any guarantee or justification that this prior is sufficiently informative or reliable? Additionally, under what assumptions does this approach provide a good approximation, and how sensitive is the method to the quality of this initial prior?

**Ethical Concerns:**

["NO or VERY MINOR ethics concerns only"]

**Final Justification:**

I would like to thank the authors for addressing many of my concerns and acknowledge that the paper has clearly improved with the additional results provided during the rebuttal. I am therefore inclined to acceptance.

**Limitations:**

The derivations and theoretical results in the paper are based entirely on linear Structural Equation Models (SEMs), which are relatively simplistic and may not adequately reflect the complexity of real-world data-generating processes. My primary concern is the practicality and robustness of the proposed algorithm in more general settings with more complex models such a generalized SEM or other nonlinear formulations that better capture real-world causal structures. It would be valuable for the authors to discuss the potential limitations of their method in such contexts, as well as the prospects for extending the framework beyond the linear case.

**Quality:**

4

**Strengths And Weaknesses:**

Strengths:
- The theoretical analysis is rigorous. The authors provide a clear and detailed mechanism for how imprecise priors can distort learned DAG structures, and justify that the error is influenced by the triangular and double collision pattern. The proposed algorithm is grounded in such an observation and has strong motivation.
- The paper is well-written and easy to follow in general.
- The experimental validation is thorough within the considered scenarios, lending credibility to the reported improvements.

Weakness
- My primary concern lies in the limited practical applicability of the theoretical framework and the proposed method. The theoretical analysis is confined to linear SEMs and relies on the faithfulness assumption—an assumption that may not hold in many real-world scenarios. For example, when analyzing the impact of edge weight deviation, the framework assumes optimization over linear SEMs, whereas in practice, structural relationships can be nonlinear or governed by more complex forms such as generalized SEMs. Similarly, the construction of the adaptive prior is based on the assumption that random DAGs follow Erdös-Rényi (ER) graphs, and key theoretical results (e.g., the lemma) depend on this assumption. These constraints narrow the scope of the method and limit the generalizability of its conclusions to broader or more realistic settings.
- The experimental validation is currently restricted to synthetic datasets generated using ER and scale-free (SF) models. While the authors explore various configurations, including different edge-to-node ratios and noise types, the evaluation lacks diversity in underlying graph models. It would strengthen the paper to test PGAP on a broader set of random graph models, especially those where the frequency of triangular and double collision patterns differs substantially from ER/SF. More experiments on real-world datasets are also recommended. Moreover, results in Appendix C suggest that PGAP yields less significant improvements in nonlinear settings, raising questions about the method’s effectiveness beyond the linear domain and further indicating that the identified patterns and deviation mechanisms may not fully capture the complexity of nonlinear models.
- I would also like to raise concerns about the method used to obtain the initial prior. According to Algorithms 1 and 2, the prior is constructed via a single round of base structure learning. It is unclear whether this process reliably yields a meaningful or accurate prior. Further clarification is needed on the conditions under which this approach can be expected to work well, and whether any theoretical guarantees or empirical justifications support it.

---

> ### Author Rebuttal · Authors · 2025-07-27
>
> We sincerely thank you for your detailed and thoughtful feedback. We will address your questions in order below and would be happy to answer any further questions you might have.
>
> **Q1:**
> Linear SEMs are a foundational tool in causal discovery, not only being used extensively across many scientific and industrial domains [1-3] but also serving as a critical object of study for the development of more advanced methods [4-6]. Although nonlinear models can often better capture the complexity of real-world relationships, the continued prevalence of linear models is largely due to their interpretability and computational tractability. The linear coefficients provide a clear measure of causal influence. In many complex systems, linear models also serve as powerful first-order approximations of the underlying relationships. This is especially true in high-dimensional, low-sample-size settings, where fitting complex nonlinear models is often statistically infeasible and highly prone to overfitting. In such scenarios, the linear assumption becomes a pragmatic and often necessary choice for obtaining robust causal structures. Given that linear SEMs are a cornerstone of causal modeling, we believe that addressing the fundamental, unresolved problem of how to robustly integrate imprecise prior knowledge within this domain is a significant contribution.
>
> **Q2:**
> Our formal theoretical analysis is developed within the linear SEM framework to allow for a precise characterization of deviation propagation. Although extending this formal analysis to general nonlinear models is a challenging task, the core principle of our PGAP framework is generalizable. We outline our path for this extension as follows.
>
> First, we generalize our core concepts from the parametric to the nonparametric domain, leveraging the framework of Zheng et al. [7]. The concept of an edge weight is replaced by a functional dependency measure, $[W(f)]_{kj}:= ||\partial _k f_j|| _{L^2}$, and "edge weight deviation" becomes a more general "functional dependency deviation."
>
> To understand how this functional deviation propagates, we next adopt a local perspective. When a prior introduces a functional deviation, the learned function $\hat{f}_j$ and the data-optimal function $f_j^{\star}$ are no longer identical. This is a critical distinction from the linear setting. In a linear model, this functional difference is inherently global; a change in a single coefficient means the two linear functions differ across the entire input space. In the nonlinear case, however, the two functions $\hat{f}_j$ and $f_j^{\star}$ can be identical over large areas and only diverge in specific local regions of the input space. It is within these regions of divergence that we can analyze the propagation mechanism.
>
> In these regions, the behavior of $\hat{f}_j$ can be approximated by its first-order Taylor expansion, rendering it locally linear. The partial derivatives of $\nabla\hat{f}_j$ act as the coefficients of this local linear approximation. The pressure from the prior forces the value of $\partial _{\beta}\hat{f}_j$ in this region to be non-optimal. To minimize the loss within this local context, the optimization process is incentivized to adjust the other "local coefficients," $\partial _{k}\hat{f}_j$ for $k \ne \beta$. This local mechanism is analogous to the error propagation we formally analyzed in the linear setting. In this case, the global functional dependency measure, $[W(f)] _{kj}$, will also become non-zero. This signifies the creation of a spurious edge $(X_k \rightarrow X_j)$ and leads to the formation of the topological patterns we leverage in PGAP.
>
> This local perspective also provides a crucial insight into why the effects of this deviation might appear less pronounced in some nonlinear settings. The global objective function is optimized over a finite set of sample points. If a local region where the prior causes significant deviation is sparsely sampled, its contribution to the global loss will be minimal. The optimizer, therefore, may find a solution that largely ignores the prior's influence in these data-scarce regions, resulting in a change to the global dependency measure $|| \partial _k \hat{f}_j || _{L^2}$ that is only minor.
>
> This does not mean the deviation phenomenon is negligible in nonlinear models. On the contrary, it points to a crucial and often overlooked aspect of causal discovery. While global dependency metrics are standard, they can average out important local effects that occur only in specific regions of the input space. This local perspective is particularly valuable for many emerging applications, such as discovering context-specific causal effects or understanding heterogeneous treatment effects in subgroups. In these scenarios, analyzing and accounting for local prior-induced deviation represents a key step forward for the field, and we believe that as the field moves towards more fine-grained functional causal analysis, the issues of local deviation we have identified will become increasingly critical to address.
>
> **Q3:**
> We thank the reviewer for pointing out this ambiguity. To be clear, the prior knowledge is an external input to our framework, not something generated by our algorithm. In line with standard practice in this research area, we assume this prior knowledge is qualitatively accurate. To strictly ensure this condition in our experiments, all priors were generated by randomly selecting true edges from the ground-truth graph. Our algorithms, therefore, describe a method for robustly using this externally-provided prior, not for generating it. We will revise our manuscript to make this distinction clear.
>
> **Regarding what is not mentioned in the question but mentioned in the weaknesses:**
> * Regarding the ER graph assumption, our core deviation mechanism (Lemmas 2-3) is entirely general. We used the ER model only for the analysis in Lemmas 4-5 to obtain a concise, closed-form expression for clarity, as a general model would yield a complex result obscuring the intuition. This simple expression allows us to clearly demonstrate our key justification: that deviation-induced patterns occur at a much higher frequency, making them a strong error signal. We will clarify this distinction in our revision.
> * In response to the reviewer's suggestion, we further validated PGAP's effectiveness on three real-world datasets of varying scales: the small-scale PaidSearch (7 nodes, 6 edges) [8], and two larger-scale datasets, Magic-niab (44 nodes, 66 edges) [9] and Ecoli70 (46 nodes, 70 edges) [10].
>
> **Table 1:** Improved effect of PGAP on the different datasets.
> | Dataset              |    PS    |          |         |         |    Mn    |          |         |         |   E70    |          |         |         |
> | :------------------- | :------: | :------: | :-----: | :-----: | :------: | :------: | :-----: | :-----: | :------: | :------: | :-----: | :-----: |
> | **Metric**           |  **F1**  |          | **SHD** |         |  **F1**  |          | **SHD** |         |  **F1**  |          | **SHD** |         |
> | **Prior Proportion** | **0.2**  | **0.4**  | **0.2** | **0.4** | **0.2**  | **0.4**  | **0.2** | **0.4** | **0.2**  | **0.4**  | **0.2** | **0.4** |
> | NOTEARS-Hard         |   15.4   |   42.9   |    9    |    7    |   34.9   |   46.7   |   51    |   45    |   26.7   |   35.0   |   123   |   112   |
> | +PGAP                |   15.4   | **46.2** |    9    |  **6**  | **35.7** | **54.2** | **50**  | **40**  | **28.4** | **38.7** | **107** | **103** |
> | NOTEARS-ReLU         |   13.3   |   30.8   |   11    |   10    |   35.3   |   42.7   |   51    |   47    |   23.0   |   29.5   |   161   |   176   |
> | +PGAP                | **16.7** | **42.9** |  **8**  |  **7**  | **37.2** | **53.2** | **49**  | **41**  | **27.9** | **35.7** | **117** | **99**  |
> | NOTEARS-Softmax      |   30.8   |   40.0   |    8    |    8    |   35.3   |   44.9   |   51    |   46    |   20.6   |   29.0   |   155   |   176   |
> | +PGAP                | **33.3** | **46.2** |  **7**  |  **6**  | **37.2** | **54.7** | **49**  | **40**  | **25.2** | **36.0** | **113** | **108** |
>
> As shown in Table 1, the results demonstrate that PGAP provides consistent and significant improvements across all three datasets. When integrated with the baseline methods, PGAP leads to better outcomes, reflected in both higher F1 scores and lower SHD values. This robust performance on datasets of different sizes and domains underscores the practical applicability and generalizability of our proposed framework.
>
> Regarding the diversity of graph models, we also conducted additional experiments. Due to space constraints in this reply, we have included the detailed results in our response to **Reviewer xCbJ**, but they are intended to directly address the concerns raised here. We believe these new results, along with our experiments on additional real-world datasets, further strengthen the case for PGAP's practical applicability and generalizability.
>
> * We thank for raising this concern and have addressed this point in Q3.
>
> [1] Detecting abnormal connectivity in schizophrenia via a joint directed acyclic graph estimation model.
>
> [2] Benchmarking of Data-Driven Causality Discovery Approaches in the Interactions of Arctic Sea Ice and Atmosphere.
>
> [3] Scalable causal structure learning: Scoping review of traditional and deep learning algorithms and new opportunities in biomedicine.
>
> [4] DYNOTEARS: Structure Learning from Time-Series Data.
>
> [5] Federated Bayesian network learning from multi-site data.
>
> [6] A hybrid constrained continuous optimization approach for optimal causal discovery from biological data.
>
> [7] Learning sparse nonparametric dags.
>
> [8] A model of individual keyword performance in paid search advertising.
>
> [9] Multiple quantitative trait analysis using Bayesian networks.
>
> [10] A shrinkage approach to large-scale covariance matrix estimation and implications for functional genomics.

---

> > ### Comment · Reviewer_7Uc5 · 2025-08-04
> >
> > Dear Authors,
> >
> > Thank you for your clarification and the additional experiments. Most of my concern has been addressed.
> >
> > I still have a minor question related to Q1, I understand that the linear SEM is the cornerstone for causal modelling, but it would be good to see the assumptions behind eq (1) and (3) so that we can better understand the applicability of linear SEM and when it fails.
> >
> > Overall, I am satisfied with the responses, and the additional experiments show the practical applicability of the framework. I will increase my score accordingly.

---

> ### Author Response · Authors · 2025-08-04
> **Response to Reviewer 7Uc5's Final Question**
>
> We sincerely thank you for your positive feedback and the insightful question regarding the model's foundational assumptions. Clarifying them is indeed crucial for understanding the applicability of our framework, and we are happy to elaborate on this below.
>
> **Assumptions for the Linear SEM (Eq. 1):**
>
> $X = B^T X + \epsilon$
>
> Equation (1) defines a linear Structural Equation Model, which rests on two key assumptions:
>
> * **Linearity:** The model assumes that the causal relationships between variables are linear, where each variable $X_i$ is a weighted linear sum of its parent variables plus an error term. This formulation serves as a powerful and interpretable foundation for causal analysis. As we discussed in our response to your Q2, the core principles of our framework can be extended to nonlinear settings, where local linear approximations can reveal similar deviation phenomena.
>
> * **Acyclicity:** The model assumes the underlying causal structure is a Directed Acyclic Graph (DAG). This is a cornerstone of causal modeling with directed graphs, as it precludes feedback loops. This assumption reflects the fundamental intuition that a cause must precede its effect. The vast majority of methods in this area operate under this fundamental assumption.
>
> **Assumptions for the Scoring Functions (Eq. 3):**
>
> The scoring functions (Eq. 3), which measure how well a candidate graph W fits the data X, are given by:
> $$Q_{ls}(W;X)=\frac{1}{2n}||XW-X||_{F}^{2}$$
>
> $$Q_{nll}(W;X)=\frac{1}{2n}\sum_{i=1}^{d}log||XW_{i}-X_{i}||_{2}^{2}$$
> The choice between these scores is tied to assumptions about the distribution of the error terms $\epsilon_i$.
>
> * **Least Squares ($Q_{ls}$):** The least squares score is equivalent to the (negative) log-likelihood of the data if we assume the error terms $\epsilon_i$ are drawn from an **Gaussian distribution with zero mean and equal variance** for all variables (i.e., $\epsilon_i \sim \mathcal{N}(0, \sigma^2)$ for all $i$).
>
> * **Negative Log-Likelihood ($Q_{nll}$):** The NLL score generalizes the least squares score by corresponding to the log-likelihood under the assumption of **Gaussian errors with zero mean but allowing for unequal variances** (i.e., $\epsilon_i \sim \mathcal{N}(0, \sigma_i^2)$).
>
> Our comprehensive experimental designs were created to test the robustness of PGAP under diverse statistical conditions, both matching and deviating from these formal assumptions. For instance, when the true noise follows an **Exponential or Gumbel distribution**, as tested in our experiments, the NLL scoring function (which assumes Gaussian noise) becomes a quasi-likelihood because its statistical assumptions are not perfectly met. The consistent performance improvements demonstrated in our experiment (not only in these non-Gaussian scenarios but also across different equal-variance (-EV) and non-equal-variance (-NV) setting) suggest that the deviation mechanism we identify is a fundamental phenomenon. This shows that PGAP is effective at mitigating these effects even when the statistical model is not perfectly specified.
>
> We hope this detailed explanation fully addresses your question and will add a summary of these key assumptions to the Appendix to improve the clarity of the final manuscript. We are confident in our proposed framework and its contributions. Should you have any other questions, we would be very happy to undertake any further work to ensure this paper meets the standards of acceptance you envision.

---

> ### Author Response · Authors · 2025-08-08
>
> We would like to once again express our sincere gratitude for your constructive feedback and insightful questions throughout the review process.
>
> Regarding your final question about the foundational assumptions of our model, we mentioned in our last response that we would add a clarification to the appendix to enhance the manuscript's clarity for future readers. To follow up on this, we have drafted the paragraph that we plan to include in the revised manuscript's appendix. We believe this addition, prompted by your valuable suggestion, significantly strengthens the paper, and we wanted to share it with you for completeness.
>
> The draft for the appendix is as follows:
>
> **Appendix X: Foundational Assumptions of the Linear SEM and Scoring Functions**
>
> The theoretical framework presented in this paper is grounded in the linear Structural Equation Model (SEM), a widely adopted model for causal discovery. The validity and interpretation of our results rest on several key assumptions, which we delineate here for clarity.
>
> **Assumptions for the Linear SEM (Eq. 1):** Our model, defined as $X = B^T X + \epsilon$, is based on two primary assumptions. First is “Linearity,'' which posits that the causal relationship between a variable $X_i$ and its parents $Pa(X_i)$ is a linear combination, weighted by the elements of the true adjacency matrix $B$. While a simplification of many real world processes, this assumption provides crucial tractability and interpretability. Second is ``Acyclicity,'' which requires the underlying graph structure to be a Directed Acyclic Graph (DAG). This is a cornerstone assumption in most graphical causal models, ensuring that a variable cannot be its own ancestor and thus precluding feedback loops.
>
> **Assumptions for the Scoring Functions (Eq. 3):** The optimization objective for learning a candidate graph $W$ relies on a scoring function, such as $Q_{ls}(W;X)$ or $Q_{nll}(W;X)$. The choice between these scores implies assumptions about the noise distribution. The least squares score, $Q_{ls}$, is statistically equivalent to the (negative) log-likelihood under the assumption of independent and identically distributed Gaussian noise terms, i.e., $\epsilon_i \sim \mathcal{N}(0, \sigma^2)$ for all $i$. The more general negative log-likelihood score, $Q_{nll}$, relaxes this by allowing for differing variances across noise terms, i.e., $\epsilon_i \sim \mathcal{N}(0, \sigma_i^2)$.
>
> Furthermore, we wish to emphasize that the core deviation propagation mechanism identified in our work appears robust even when these statistical assumptions are not perfectly met. As demonstrated in our experiments with non-Gaussian noise (e.g., Exponential, Gumbel), PGAP consistently provides improvements. This suggests that the identified triangular and double collision patterns are fundamental indicators of prior-induced error, and their utility is not strictly confined to scenarios where the model is perfectly specified. This robustness is a key strength of our proposed framework.
>
> Thank you again for your time and for guiding us to improve the paper. We are confident that these additions will be very helpful for the reader.

---

### Decision · Program_Chairs · 2025-09-17

**Decision:**

Accept (poster)

**Comment:**

The paper analyzes how imprecise edge priors distort continuous DAG learning in linear SEMs, proving that prior-induced weight deviations propagate as first-order errors that manifest exactly as triangles and double collisions; it then introduces PGAP, a lightweight wrapper that detects these motifs after an initial run and adaptively down-weights the offending priors before re-optimization. After rebuttal, all reviewers lean accept, noting clearer statement of assumptions (linearity, acyclicity, noise models), stronger justification that the two motifs are exhaustive for first-order propagation, added real-data results (PaidSearch, Magic-niab, Ecoli70) and diverse graph families (WS, SBM, GRG), plus a data-driven tuning scheme that stabilizes performance on dense graphs; clarifications on “accurate vs. precise” priors and practical guidance also helped. The remaining limits are that theory stays in the linear/faithful setting, PGAP has no formal consistency guarantees, and dense-graph gains are more sensitive (though mitigated).